# EXPLAINABLE EVIDENTIAL CLUSTERING

## ABSTRACT

Unsupervised classification is a core problem in machine learning. Because real-world data are often imperfect, non-additive frameworks, such as evidential clustering, grounded in Dempster-Shafer theory, explicitly handle uncertainty and imprecision. These frameworks are particularly well suited to high-stakes decisions, which tend to require both interpretability and cautiousness. However, while decision-tree surrogates have enabled transparent explanations for hard clustering, explainability for evidential clustering remains largely unexplored. We address this gap by formalizing representativeness, a utility-based criterion that captures decision-makers' preferences over explanation misassignments, and introducing evidential mistakeness, a loss function tailored to credal partitions. Building on these foundations, we propose the Iterative Evidential Mistakeness Minimization (IEMM) algorithm, which learns decision-tree explainers for evidential clustering by optimizing representativeness under uncertainty and imprecision. We provide theoretical conditions for effective explanations in both hard and evidential settings and show how utility function parameters can be set to reflect different decision attitudes. Experiments on synthetic and real-world datasets demonstrate that IEMM improves the performance of existing methods by producing representative and preference-aligned explanations of evidential clusterings, supporting cautious, transparent analysis in the presence of imperfect data.

## 1 INTRODUCTION

**Clustering** is a fundamental machine learning problem MacQueen (1967) that aims to group similar objects while distinguishing different ones Hansen & Jaumard (1997). As a core data analysis task, it reveals patterns and enables applications such as data compression, summarization, visualization, and anomaly detection Xu & Wunsch (2005). As with other machine learning methods, two major challenges persist in clustering: **imperfections in the input data** Hüllermeier & Waegeman (2021) and **interpretability** Carvalho et al. (2019).

Real-world scenarios with imperfect data require **cautiousness** Bengs et al. (2022); Angelopoulos et al. (2022); Imoussaten & Jacquin (2022); Hüllermeier et al. (2022); Nguyen et al. (2018), defined as decision-makers' awareness of model limitations and resulting risk-aversion. Effective cautiousness depends on properly characterizing these imperfections, primarily **uncertainty** and **imprecision** Dubois & Prade (2009). In machine learning contexts, imperfections typically arise from weak supervision, aleatoric uncertainty (intrinsic variability in the data), and epistemic uncertainty (a lack of data in parts of the feature space) Hüllermeier & Waegeman (2021). Approaches that address these issues include imprecise probability theory Walley (1991), possibility theory Dubois & Prade (1988), rough sets Pawlak (1982), fuzzy sets Zadeh (1965), and Dempster-Shafer evidence theory Shafer (1976).

These foundations have given rise to various clustering methods, including fuzzy Ruspini (1969), possibilistic Krishnapuram & Keller (1993), rough Lingras & West (2004), and evidential clustering Masson & Denœux (2008). In the latter framework, while classical hard clustering Hartigan & Wong (1979) assigns each point to exactly one cluster, evidential clustering induces a credal partition Masson & Denœux (2008) that represents both uncertainty and imprecision through partial membership across multiple cluster combinations.

Interpreting clustering results is equally critical: without interpretation, clusterings often lack practical utility. This has motivated a growing body of work on interpretable clustering Ben-Hur et al.

(2001); Alvarez-Garcia et al. (2024); Tutay & Somech (2023); Carrizosa et al. (2022); Lawless & Gunluk (2022). A prominent approach borrows from supervised learning: treat the hard clustering labels as ground truth and train a surrogate decision-tree classifier to reproduce them. A notable example is Iterative Mistake Minimization (IMM) Moshkovitz et al. (2020), which fits a decision tree aligned with the centroid structure—each leaf holds exactly one centroid, mapping its points to the associated cluster and offering interpretability guarantees regarding $k$-means and $k$-medians clustering objectives. Building on the IMM, subsequent work strengthens theoretical guarantees of the provided explanation and broadens its scope to other methods. Many of these rely on randomization to improve worst-case bounds Makarychev & Shan (2022; 2023); Esfandiari et al. (2021); Bandyapadhyay et al. (2023), while others explore shallow decision trees, with the same numbers of leaves but not necessarily aligned with centroids Laber et al. (2022); Frost et al. (2020); Makarychev & Shan (2022). Some other works investigate different clustering objectives Laber & Murtinho (2021); Gamlath et al. (2021); Fleissner et al. (2024) or introducing oblique decision trees Gabidolla & Carreira-Perpiñán (2022).

**Explainability** refers to a model's ability to provide clear, audience-appropriate reasons for its behavior Barredo Arrieta et al. (2020). It enables users to understand, critique, and improve models. A common taxonomy distinguishes **intrinsic** methods—models designed to be interpretable—from **post-hoc** methods—explanations for already trained black boxes Carvalho et al. (2019). Post-hoc techniques mainly fall into two families Barredo Arrieta et al. (2020): (i) feature-relevance methods, which rank or quantify the influence of input features Lundberg et al. (2019); Baehrens et al. (2010) but, because they reveal little about the dataset's structure Moshkovitz et al. (2020), face criticism in high-stakes settings Rudin (2019); and (ii) **simplification** methods, which approximate black-box classifiers with interpretable surrogates, such as decision trees, rule lists, or linear models Guidotti et al. (2018). These categories are not mutually exclusive: explanations may also be delivered through examples, counterfactuals, or visual/textual modalities Barredo Arrieta et al. (2020). Nor are they exhaustive. For example, feature relevance can be obtained via simplification, as in LIME Ribeiro et al. (2016). Our proposed approach can be viewed as both intrinsic (it produces interpretable models) and post-hoc simplification (it explains a given clustering).

High-stakes domains such as healthcare demand both interpretability and cautiousness. Yet only a few works extend explainability to imprecise methods. Early efforts focus on evidential supervised classification via counterfactuals and feature importance Zhang (2023). Similar methods have been applied to fuzzy or possibilistic clustering Ellis et al. (2021; 2024); Baaj (2022). Other works develop intrinsically interpretable non-hard clustering methods Adamyan et al. (2025); Jiao et al. (2022; 2023). However, these approaches do not provide post-hoc explanations for existing evidential clustering functions obtained by other methods. To our knowledge, no prior work explains evidential clustering in this manner. As noted in literature Zhang et al. (2024), explainable clustering over uncertain or imprecise data warrants investigation to enable cautious, transparent analysis under imperfect data sources. This paper addresses that gap.

The main **objectives** of this paper are:

1. To conduct a comprehensive investigation of decision trees as explainers for hard clustering functions, establishing conditions that define effective explanations.

2. To develop a theoretical framework that extends these conditions to encompass uncertainty and imprecision, particularly within the evidential clustering paradigm.

3. To introduce an innovative **Explainable Evidential Clustering** method through a novel algorithm grounded in these theoretical foundations.

Our key **contributions** include:

1. We demonstrate that representativity is a necessary and sufficient condition for decision trees to act as abductive explainers in the hard case. Building upon utility functions, we introduce the concept of *Evidential Representativeness*, which quantifies decision-makers' preferences regarding errors committed by an explainer. This advancement enables systematic evaluation of cautious explanations.

2. We propose the Evidential Mistakeness function, demonstrate that minimizing it leads to representative explanations, and develop the **Iterative Evidential Mistakeness Minimization**

(IEMM) algorithm. This novel approach, inspired by the IMM algorithm Moshkovitz et al. (2020), generates surrogate decision trees that effectively explain evidential clustering functions.

3. We implement and validate this algorithm on both synthetic and real-world datasets, demonstrating how to select utility parameters that reflect different decision attitudes.

The remainder of this paper is structured as follows. Section 2 presents the theoretical foundations, encompassing belief functions, evidential clustering, and explainability. Section 3 introduces the concepts on which our approach is based: a specific family of utility functions, the representativeness criterion, the Evidential Mistakeness loss function, and the IEMM algorithm. We provide formal analysis of these concepts along with illustrative examples. We also make available the IEMM Python package and the complete code for all experiments at OMITTED TO AVOID IDENTIFICATION.

## 2 BACKGROUND

Let $X$ represent a set of **observations** in a known **feature space** $\mathbb{X}$. We assume $X = \{x_1, ..., x_N\} \subset \mathbb{X} = \mathcal{A}_1 \times \ldots \times \mathcal{A}_D$, where each element of $\mathcal{D} = \{\mathcal{A}_1, \ldots, \mathcal{A}_D\}$ is called an **attribute**. We assume all attributes are finite[1]. In essence, $X$ is a set of $D$ measurements for each of $N$ objects, while $\mathbb{X}$ encompasses all possible measurements.

A classification problem is the task of assigning each observation in $X$ to an outcome from a finite set $\Omega = \{\omega_1, ..., \omega_C\}$, which we call the **frame of discernment**. A function that performs this assignment is called a **classifier**. When a set of training examples is available, we refer to the problem of constructing such function as **supervised classification**. In contrast, if no training examples are available and the goal is to group observations based on their similarity—without prior knowledge of the classes or labels—the task is called **unsupervised classification** or **clustering**.

### 2.1 BELIEF FUNCTIONS

The Dempster-Shafer theory of evidence Shafer (1976) provides a framework for representing uncertain and imprecise information. At the core of this theory lies the **mass of belief function**, or simply **mass function**—a map defined as:

$$m : 2^\Omega \to [0, 1] \text{ such that } \sum_{A \subseteq \Omega} m(A) = 1.$$

Within this framework, an element $\omega \in \Omega$ represents the finest level of discernible information. The mass $m(A)$ quantifies the degree of confidence in the statement that 'the correct hypothesis $\omega$ belongs to $A \subseteq \Omega$, yet it remains impossible to determine which specific element of $A$ is correct'. When $m(\varnothing) = 0$, we say the mass *satisfies the closed-world hypothesis* Smets (1988), meaning it rejects the possibility that the correct hypothesis $\omega$ lies outside $\Omega$.

We define the **focal set** of $m$ as $\mathbb{F}_m = m^{-1}(]0, 1])$—the collection of all subsets of $\Omega$ assigned nonzero belief. Each member of this set is known as a **focal element**. Mass functions can be categorized based on their focal elements:

- If all focal elements are singletons (of cardinality 1), then $p(\omega) = m(\{\omega\})$ forms a probability mass function, and $m$ is called a Bayesian mass function.
- A mass function with exactly one focal element $A$ is called *categorical*, representing the logical assertion that '$\omega$ belongs to $A$'. If this single focal element is $\Omega$ itself, the function is *vacuous*, conveying no information beyond the closed-world hypothesis.

We denote by $\mathbb{M}$ the set of all mass functions defined on $\Omega$. For notational simplicity, we may write $\omega_i \cup \omega_j \cup \omega_k \cup ...$ to represent the subset $\{\omega_i, \omega_j, \omega_k, ...\}$. In Appendix A we present some additional constructions (belief/plausability/pignistic functions) allowing decision-making for a mass of belief function.

---

[1]This finite attribute assumption is crucial for decision tree operations. This assumption reflects this work's aim of explaining the clustering of tabular data. We may occasionally refer to $\mathbb{X}$ as $\mathbb{R}^D$ for simplicity, though this is an abuse of notation. When working with continuous attributes, we implicitly discretize the space (for example, with a dataset $X \subset \mathbb{R}^D$, we typically consider binary attributes $\mathcal{A}_{d,\theta} = \{True, False\}$ for each dimension $d \in \{1, ..., D\}$ and threshold $\theta \in \{x_d : x \in X\}$, where $x_{\mathcal{A}_{d,\theta}} = True$ if and only if $x_d \geq \theta$).

## 2.2 Evidential Clustering

**Definition 1.** An **evidential clustering** is a map $\mathcal{M} : X \to \mathbb{M}$.

For an observation $x \in X$, the function $\mathcal{M}(x)$, which we may denote as $m_x$, when evaluated at $A \subseteq \Omega$, returns the degree of confidence attributed to the statement 'the class $\omega$ corresponding to $x$ belongs to $A$, and it is not possible, given the available information, to determine which specific element of $A$ is the correct one'.

We refer to each element of $\Omega$ as a cluster and each subset of $\Omega$ as a metacluster. Within the context of an evidential clustering function, we denote $\mathbb{F}_\mathcal{M} = \bigcup_{x \in X} \mathbb{F}_{m_x}$. When all $m_x$ are categorical, we say that $\mathcal{M}$ is *categorical*. Similarly, when all elements of $\mathbb{F}_\mathcal{M}$ are singletons, we call $\mathcal{M}$ *bayesian*. An evidential clustering function that is both categorical and bayesian naturally induces a hard clustering. A **hard clustering** is simply a partition of observations into clusters, formalized as a surjection $\mathcal{C} : X \to \Omega$. Appendix A further discusses evidential clustering and offers a visualization. Some clustering algorithms naturally produce a centroid for each cluster. A **centroid** $v_\omega$ (resp. $v_A$) is a point in the feature space $\mathbb{X}$ that represents its cluster $\omega \in \Omega$ (resp. metacluster $A \subset \Omega$). For the remainder of this work, we assume $\varnothing \notin \mathbb{F}_\mathcal{M}$, rejecting the outlier hypothesis.

## 2.3 Decision Trees as Explainers

**Decision Trees** (DTs) Quinlan (1987) are classical machine learning algorithms, classifiers based on rooted computation trees expressed as recursive partitions of the observation space Rokach & Maimon (2005). Building upon these partitional aspects, a **node** of a decision tree is defined as the subset $S \subseteq \mathbb{X}$ associated with its vertices. Decision trees are widely used in explainability for their inherent interpretability Barredo Arrieta et al. (2020), as they yield "a set of decision rules with the if–then form" Guidotti et al. (2018). In this context, a particularly desirable outcome Amgoud & Ben-Naim (2022) is an abductive explanation: it answers the question "Why is $\Gamma(x) = \omega$?" by providing a sufficient reason for assigning the label $\omega$, where $\Gamma : \mathbb{X} \to \Omega$ is a supervised classifier Ignatiev et al. (2019). Throughout the paper, we denote by **C** the set of all consistent subsets of feature literals[2] and call an explainer any map $\chi_\Gamma : \Omega \to 2^{\mathbf{C}}$. Further definitions and formal aspects of explainers are provided in Appendix B.

A decision tree $\Delta : \mathbb{X} \to \Omega$ induces an explainer $\chi_\Gamma^\Delta : \Omega \to 2^{\mathbf{C}}$ that provides abductive explanations to a supervised classifier $\Gamma$ if and only if $\Delta = \Gamma$. This equality between the original classifier and the DT surrogate model is equivalent to the property that Amgoud & Ben-Naim (2022) calls representativity. A representative explainer is one that, for all observations $x$ with label $\omega = \Gamma(x)$, can provide an explanation in $\chi_\Gamma^\Delta(\omega)$ that holds at $x$. More details on the DT construction from the standpoint of explainability can be found in appendix C, along with our proposed proof that representativity is a necessary and sufficient condition for decision trees to provide abductive explanations.

As the complete representativity is rarely attainable, it is natural to assess the quality of explanations by the "representativeness" of the explainer $\chi_\Gamma^\Delta$. This assessment, in the supervised case, is typically performed by measuring the accuracy Ribeiro et al. (2016); Izza et al. (2022b); Narodytska et al. (2019) of the underlying classifier $\Delta$. Thus, the quality of the explanation provided by $\chi_\Gamma^\Delta$ about $\Gamma$ is quantified as:

$$\text{Accuracy}_\Gamma(\Delta) = \frac{|\{x \in X : \Gamma(x) = \Delta(x)\}|}{|X|}. \tag{1}$$

### 2.3.1 The IMM: Decision Trees Explaining Hard Clustering

Iterative Mistake Minimization (IMM) Moshkovitz et al. (2020) explains a hard clustering by training a decision tree that mimics the cluster assignments (see Figure 1) while minimizing the price of explanation. To this end, it relies on the concept of *mistake*.

**Definition 2.** Let $\mathcal{C}$ be a hard clustering function. A **mistake** in a decision-tree (DT) node $S \subseteq \mathbb{X}$ occurs when a point $x \in S$ has its associated cluster centroid $v_{\mathcal{C}(x)}$ outside of $S$, i.e., $v_{\mathcal{C}(x)} \notin S$.

---

[2]A feature literal is a pair $\langle \mathcal{A}, v \rangle$ where $\mathcal{A} \in \mathcal{D}$ and $v \in \mathcal{A}$. A consistent subset of feature literals is some set $L$ of feature literals such that $\langle \mathcal{A}, v \rangle, \langle \mathcal{A}, v' \rangle \in L \Rightarrow v = v'$.

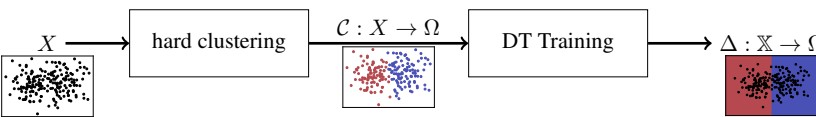

Figure 1: Schematic of explainable clustering.

The **number of mistakes** in a decision tree is the sum of mistakes committed with respect to $\mathcal{C}$ across all leaves. The IMM aims to greedily minimize the number of mistakes induced by each axis-aligned split in the decision tree Moshkovitz et al. (2020).

To assess explanation quality, as discussed in the previous section, it is natural to measure how representative the resulting explainer is. The original IMM evaluation Moshkovitz et al. (2020) relied on an explanation cost derived from the k-means and k-medians objectives. Some works Fleissner et al. (2024); Lawless & Gunluk (2022)—not restricted to these clustering methods—use the Rand index Rand (1971) or the $\text{Accuracy}_{\mathcal{C}}(\Delta)$ to assess the similarity between the original clustering and the clustering induced by the decision tree.

## 3 AN ALGORITHM FOR EXPLAINING EVIDENTIAL CLUSTERING

Our objective is to extend the concept of decision trees for cluster explanation to the evidential setting. We aim to construct a decision tree that provides a representative approximation of the evidential clustering function, as illustrated in Figure 2.

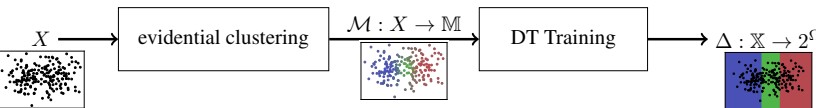

Figure 2: Scheme of Explainable Evidential Clustering.

To achieve this, we build on the hard case: we first generalize what it means for an explainer to be representative, and then, inspired by IMM, derive the notion of a mistake for evidential classifiers and propose an algorithm that seeks to minimize the resulting loss. We start with the specific case of categorical mass functions and then extend to the general case.

### 3.1 EXPLAINING CLASSIFIERS UNDER UNCERTAINTY AND IMPRECISION

Given an evidential classifier[3] $\mathcal{M} : X \to \mathbb{M}$, we seek to construct an interpretable decision tree $\Delta : \mathbb{X} \to 2^{\Omega}$ that approximates $\mathcal{M}$. We first formalize this approximation based on the quality of the generated explanations.

#### 3.1.1 UTILITY FUNCTIONS

Let us consider a categorical evidential classifier $\mathcal{M}_c : X \to \mathbb{M}$. Let us define $\overline{\mathcal{M}}_c : X \to 2^{\Omega}$ such that, for any $x \in X$ and $A \subseteq \Omega$, $\mathcal{M}_c(x)(A) = 1$ if and only if $\overline{\mathcal{M}}_c(x) = A$.

Assessing the quality of a surrogate partition $\Delta$ is challenging in the evidential setting: predictions and truths are subsets of $\Omega$, so errors vary in severity. For example, predicting $\{\omega_1, \omega_2\}$ when $\overline{\mathcal{M}}_c(x) = \{\omega_1\}$ is arguably less severe than predicting $\{\omega_2\}$, yet exact-match representativeness treats both equally. Such binary criteria ignore partial agreement and domain-specific preferences across clusters. To make these trade-offs explicit, we introduce a specific family of bounded utility functions that quantifies the decision-maker's satisfaction when $A$ is predicted while the ground truth is $B$.

**Definition 3.** A **utility function** is a map $\mathcal{U} : 2^{\Omega} \times 2^{\Omega} \to [0, 1]$ such that, $\forall A, B \in 2^{\Omega}$,

a) $\mathcal{U}(A, A) = 1$ and
b) $A \cap B = \varnothing \Rightarrow \mathcal{U}(A, B) = 0$.

---

[3]We use the term classifier to emphasize that the development of this section is valid not only for clustering but also for all evidential partitions of the data.

Utility functions are standard tools in decision theory Keeney & Raiffa (1993) and the existence of utility functions that encode decision-maker's (DM's) preferences has been widely discussed Von Neumann & Morgenstern (1947); Savage (1954); Schmeidler (1989). In the evidential setting, several works explore desirable properties and elicitation procedures for utilities. Some of them focused on the case where ground truth is assumed to be known precisely Zaffalon et al. (2012); Ma & Denœux (2021) and others Jacquin et al. (2019); Imoussaten & Jacquin (2022); Imoussaten (2023) extend to the cases where ground truth is itself imprecise. In our formulation, we consider a bounded utility as described by conditions a) and b) in Definition 3: a perfect prediction yields total satisfaction for the DM (utility equals 1), whereas completely disjoint prediction and truth yield unacceptable satisfaction for the DM (utility equals 0). This assumption is convenient for our analysis but is not the general case Kunitomo-Jacquin et al. (2025). In Appendix D, we discuss how such utilities relate to explanation costs of misassignments in the categorical case and provide an illustrative example.

### 3.1.2 EVIDENTIAL REPRESENTATIVENESS

**Definition 4.** A **cautious explainer** for a categorical evidential classifier $\mathcal{M}_c : X \to \mathbb{M}$ is a map $\chi_{\mathcal{M}_c} : 2^\Omega \to 2^{\mathbf{C}}$.

The cautious explainer differs from a standard explainer in its capability to explain the classifier's imprecise predictions. In this context, a good explainer should ensure that if $x$ is assigned to $A$ by the classifier, then there exists some $B$ such that $\mathcal{U}(A, B) = 1$ and $x$ is explained by $\chi_{\mathcal{M}_c}(B)$. We therefore characterize the representativeness of a cautious explainer with respect to a utility.

**Definition 5.** A $\mathcal{U}$-**representative cautious explainer** for a categorical evidential clustering is a cautious explainer $\chi_{\mathcal{M}_c}$ such that, $\forall A \in 2^\Omega$, $\forall x \in \overline{\mathcal{M}}_c^{-1}(\{A\})$, there exists $L \in \bigcup_{\mathcal{U}(B,A)=1} \chi_{\mathcal{M}_c}(B)$ such that, $\forall \langle \mathcal{A}, v \rangle \in L$, $x_{\mathcal{A}} = v$.

The value of defining $\mathcal{U}$-representative cautious explainers lies in the fact that, in the case of decision trees, this notion generalizes the concept of abductive explanations. In other words, for any decision tree $\Delta : \mathbb{X} \to 2^\Omega$, we can guarantee that if its induced explainer is $\mathcal{U}$-representative, then it provides a sufficient explanation for maximizing utility; i.e., $\forall A \subseteq \Omega$, if $L \in \bigcup_{\mathcal{U}(B,A)=1} \chi_{\mathcal{M}_c}^{\Delta}(B)$,

$$\left( \bigwedge_{\langle \mathcal{A}, v \rangle \in L} (x_{\mathcal{A}} = v) \right) \Rightarrow \mathcal{U}(\overline{\mathcal{M}}_c(x), A) = 1.$$

Different utility functions induce different notions of representativeness. Intuitively, more permissive utilities yield higher scores and tolerate more error. In the categorical evidential case, utility allows us to define the $\mathcal{U}$-**(categorical) representativeness** of a cautious explainer as its average utility:

$$\mathcal{R}_{\mathcal{M}_c, \mathcal{U}}(\Delta) = \frac{1}{|X|} \sum_{x \in X} \mathcal{U}(\Delta(x), \overline{\mathcal{M}}_c(x)). \tag{2}$$

The $\mathcal{U}$-representativeness of a DT equals 1 if and only if its induced explainer is $\mathcal{U}$-representative.

For any evidential partition $\mathcal{M} : X \to \mathbb{M}$, we can extend Equation 2. Let us define then the $\mathcal{U}$-**(evidential) representativeness** $\mathcal{R}_{\mathcal{M}, \mathcal{U}} : (\mathbb{X} \to 2^\Omega) \to [0, 1]$ of a cautious explainer as the expected categorical representativeness weighted by the mass function:

$$\mathcal{R}_{\mathcal{M}, \mathcal{U}}(\Delta) = \frac{1}{|X|} \sum_{x \in X} \sum_{B \in \mathbb{F}_{\mathcal{M}}} \mathcal{U}(\Delta(x), B) m_x(B). \tag{3}$$

Equation (2) is clearly a special case of (3) when $\mathcal{M}$ is categorical. Additionally, equations (3) and (1) coincide when $\mathcal{M}$ is a hard partition and $\mathcal{U}(A, B) = \mathbb{1}_{A=B}$.

The literature offers several ways to compare evidential partitions, mainly via distances between mass functions or by aggregating nonspecificity with measures of conflict Jousselme et al. (2001); Jousselme & Maupin (2012); Hoarau et al. (2023a); Denoux et al. (2018); Campagner et al. (2023); Masson & Denœux (2008). However, we believe that utility-based representativeness offers advantages for our setting. As we face an explanation task, the choice of utilities lets us encode

decision-maker preferences in a more interpretable way than tuning parameters of a clustering objective Masson & Denœux (2008) or weighting nonspecificity and conflict Denoux et al. (2018); Denoeux & Bjanger (2000), as those values often lack an immediate meaning to the explanation audience. Moreover, unlike distances, utilities can capture complex and possibly asymmetric preferences. For instance, mapping $A = \{\omega_1\}$ to $B = \{\omega_1, \omega_2\}$ need not be penalized the same as mapping $A = \{\omega_1, \omega_2\}$ to $B = \{\omega_1\}$.

### 3.1.3 EVIDENTIAL MISTAKENESS

With this updated notion of representativeness, we can extend the concept of a mistake to the evidential case as a cost function capturing the representativeness loss associated with a single DT explainer node. Recall from Definition 2 that, in the hard case, the number of mistakes in a node $S$ can be described in two equivalent ways:

1. The number of mistakes in $S$ is the number of points $x \in S$ such that $\exists v_\omega \notin S$ with $\omega = \mathcal{C}(x)$.
2. The number of mistakes in $S$ is the number of points $x \in S$ such that $\forall v_\omega \in S, \omega \neq \mathcal{C}(x)$.

Translating these to the evidential setting yields two natural definitions of **evidential mistakeness**:

1. The evidential mistakeness in $S$ is the sum of the costs introduced by not assigning points $x$ in $S$ to metaclusters that are not in $S$:

$$\overline{M}_{\mathcal{M},\mathcal{U}}(S) = \sum_{x \in S} \sum_{v_A \notin S} \sum_{B \in \mathbb{F}_{\mathcal{M}}} \mathcal{U}(A, B)m_x(B). \tag{4}$$

2. The evidential mistakeness in $S$ is the sum, over all $x$ in $S$, of the expected cost of assigning $x$ to some metacluster in $S$:

$$\underline{M}_{\mathcal{M},\mathcal{U}}(S) = \sum_{x \in S} \sum_{v_A \in S} \sum_{B \in \mathbb{F}_{\mathcal{M}}} \frac{(1 - \mathcal{U}(A, B))m_x(B)}{|\{C \in \mathbb{F}_{\mathcal{M}} : v_C \in S\}|} \tag{5}$$

One can interpret Equation (4) as the satisfaction that the DM will not concretize and Equation (5) as the unsatisfaction that the DM will concretize. When $\mathcal{M}$ induces a hard clustering and $\mathcal{U}(A, B) = \mathbb{1}_{A=B}$, Equation (4) equals the number of mistakes in $S$. Additionally, the total cost of a cautious DT explainer induced by Equation (5) is zero if and only if the explainer is $\mathcal{U}$-representative.

For IMM-like algorithms where all leaves $S$ contain exactly one centroid, both evidential mistakeness forms from Equations (4) and (5) are minimized by explanations with maximal evidential representativeness (see proof in Appendix E). The key difference between these definitions emerges in nodes containing multiple centroids, where Equation (5) penalizes such nodes more heavily than Equation (4). This makes Equation (4) better suited for conservative explainers, while Equation (5) is preferable for more risk-attractive ones.

### 3.2 THE ALGORITHM

Inspired by IMM, we propose the **Iterative Evidential Mistakeness Minimization (IEMM)**. The IEMM fits a decision tree based on an evidential clustering by minimizing the evidential mistakeness function (see Algorithm 1). Each iteration of IEMM, for a region $S \subseteq \mathbb{X}$, considers a subset $F \subseteq \mathbb{F}_{\mathcal{M}}$ of the focal sets whose metacluster centroids lie within $S$ and finds the split that, by separating centroids, minimizes the contribution to the evidential mistakeness.

From a computational perspective, the baseline IMM algorithm has complexity $O(C \cdot D \cdot N \cdot \log N)$ Moshkovitz et al. (2020), where $C = |\Omega|$ represents the number of clusters, $D$ the dimensionality of the feature space, and $N$ the sample count. Our IEMM algorithm extends this by incorporating utility computations at each node, adding an $O(K^2)$ factor where $K = |\mathbb{F}|$ is the number of metaclusters. This results in a total complexity of $O(K^2 \cdot D \cdot N \cdot \log N)$. In practice, explainable decision trees typically employ a modest number of metaclusters, mitigating potential performance concerns.

For the categorical evidential clustering setting, when the utility function $\mathcal{U}(A, B) = \mathbb{1}_{A=B}$ is used, IEMM behaves identically to IMM over the metaclusters. This allows us to adapt the structural bounds from Moshkovitz et al. (2020) to the evidential case. Specifically, we can relate the explanation cost of the decision tree produced by IEMM to that of the input Evidential C-Means clustering

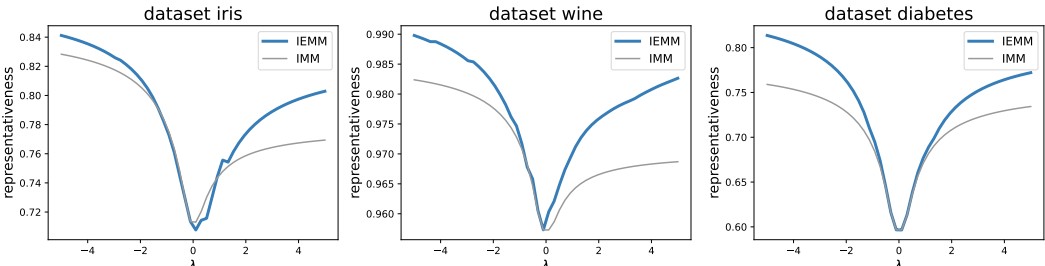

Figure 3: Obtained $\mathcal{U}^\lambda$-evidential representativeness $\mathcal{R}_{\mathcal{M},\mathcal{U}^\lambda}(\Delta)$ when explaining an ECM clustering across multiple datasets. For each dataset, we run ECM Masson & Denœux (2008) to obtain an evidential clustering function $\mathcal{M}$ and compare, for various $\lambda$, the $\Delta_{\text{IEMM}}$ learned under mistakeness $M_{\mathcal{M}}^\lambda$ with $\Delta_{\text{IMM}}$, produced by applying the adapted baseline IMM Moshkovitz et al. (2020).

Masson & Denœux (2008). This yields the bound $\mathcal{J}_{ECM}(\Delta) \leq |\Omega|^\alpha(2 + 8H|\mathbb{F}_{\overline{\mathcal{M}}_c}|) \cdot \mathcal{J}_{ECM}(\overline{\mathcal{M}}_c)$, where $\Delta$ is the IEMM decision tree and $\overline{\mathcal{M}}_c$ is the input evidential clustering. The proof and details are provided in Appendix G.

---

**Algorithm 1 IEMM**

---

**Input:** Observations $X = [x^1, \ldots, x^N] \subset \mathbb{R}^D$.
Some evidential clustering $\mathcal{M} : X \to \mathbb{M}$.
The focal sets $\mathbb{F} = \{A_1, \ldots, A_{|\mathbb{F}|}\}$
and their centroids $v = \{v^1, \ldots, v^{|\mathbb{F}|}\} \subset \mathbb{R}^D$.
**Parameter:** The chosen evidential mistakeness $\mathbf{M}$
**Output:** A decision tree $\Delta : \mathbb{R}^D \to 2^\Omega$.

1: $\Delta \leftarrow$ **split_tree**$(X, \mathcal{M}(X), \mathbb{F}, v)$
2: **function** SPLIT_TREE$(\{x^j\}_{j=1}^n, \{m^j\}_{j=1}^n, F, \{v^j\}_{A_j \in F})$
3:     **if** $|F| = 1$ **then**
4:         **leaf.metacluster** $\leftarrow F$
5:         **return leaf**
6:     **end if**
7:     **for all** $i \in [1, \ldots, D]$ **do**
8:         $\ell_i \leftarrow \min_{A_j \in F} v_i^j$
9:         $r_i \leftarrow \max_{A_j \in F} v_i^j$
10:    **end for**
11:    $i, \theta \leftarrow \arg\min_{i, \ell_i \leq \theta < r_i} \mathbf{M}(x, m, v, F, i, \theta)$
12:    $L \leftarrow \{j \mid (x_i^j \leq \theta)\}_{j=1}^n$
13:    $R \leftarrow \{j \mid (x_i^j > \theta)\}_{j=1}^n$
14:    $F_L \leftarrow \{A_j \in F \mid v_i^j \leq \theta\}$
15:    $F_R \leftarrow \{A_j \in F \mid v_i^j > \theta\}$
16:    **node.condition** $\leftarrow "x_i \leq \theta"$
17:    **node.lt** $\leftarrow$ **split_tree**$(\{x^j\}_{j\in L}, \{m^j\}_{j\in L}, F_L, v)$
18:    **node.rt** $\leftarrow$ **split_tree**$(\{x^j\}_{j\in R}, \{m^j\}_{j\in R}, F_R, v)$
19:    **return node**
20: **end function**

---

To illustrate the behavior of IEMM under different cautiousness preferences, we provide an example of usage in Appendix H, demonstrating its ability to generate decision-tree explainers that reflect varying levels of cautiousness based on the chosen utility function.

### 3.2.1 EXPERIMENTS

We evaluate whether IEMM produces explanations that align with a decision-maker's (DM's) preference for cautiousness. Preferences are modeled via a utility function family $\{\mathcal{U}^\lambda\}_{\lambda \in \mathbb{R} \cup \{\pm\infty\}}$, introduced in Appendix F, which captures different risk attitudes: larger $\lambda$ values correspond to more cautious choices, trading conflict for non-specificity in the spirit of Denoux et al. (2018). In practical applications, the utility should be elicited directly with the DM Kunitomo-Jacquin et al. (2025); here we vary $\lambda$ to study behavior across a spectrum of attitudes.

The $\lambda$-evidential mistakeness function $M_{\mathcal{M}}^\lambda$ (as described in Appendix F) is designed to be consistent with this utility family. At $\lambda = 0$, all errors are weighted equally, recovering the original IMM mistakeness in the hard-clustering case.

Because IEMM is, to our knowledge, the first algorithm able to explain cautious partitions, we compare it against a careful adaptation of IMM Moshkovitz et al. (2020). Given an evidential

clustering function $\mathcal{M}$, we derive a categorical partition by applying the strong dominance criterion pointwise (Appendix A), and then run IMM while treating each metacluster as an ordinary cluster.

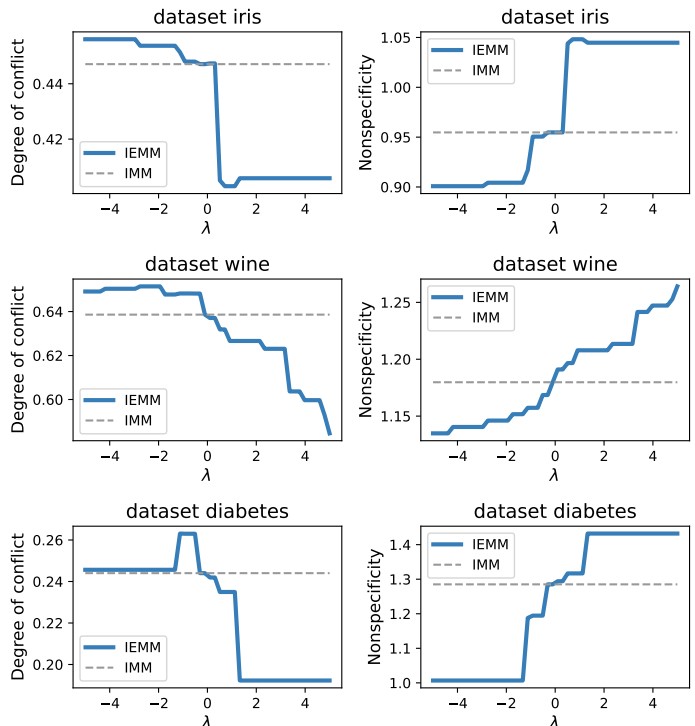

Figure 4: Conflict/non-specificity analysis for IEMM and IMM across $\lambda$ values. Higher $\lambda$ yields more cautious (less conflicting but more non-specific) explanations; lower $\lambda$ emphasizes specificity. Metrics are computed following Denoux et al. (2018).

**Experimental protocol:** For each dataset (Iris Fisher (1936), Wine Aeberhard et al. (1991), and Diabetes Efron et al. (2004) from `sklearn` Pedregosa et al. (2011)), we first run ECM Masson & Denœux (2008) to obtain $\mathcal{M}$. For a grid of $\lambda$ values, we learn $\Delta_{\text{IEMM}}$ by minimizing $M_{\mathcal{M}}^{\lambda}$ and obtain $\Delta_{\text{IMM}}$ from the induced categorical partition. We then measure $\mathcal{U}^{\lambda}$-evidential representativeness $\mathcal{R}_{\mathcal{M}, \mathcal{U}^{\lambda}}(\Delta)$ for both explainers. Results are summarized in Figure 3.

**Main findings:** IEMM consistently achieves higher $\mathcal{U}^{\lambda}$-evidential representativeness than IMM, with the difference vanishing at $\lambda = 0$. This indicates that IEMM better preserves the cautiousness inherent in the evidential clustering, while remaining competitive when cautiousness is not emphasized. This highlights the IEMM's ability to adapt explanations to the DM's risk attitude, effectively balancing specificity and cautiousness as desired, while IMM lacks this flexibility.

**Trade-off analysis:** Conflict and non-specificity define antagonistic objectives in evidential clustering Denoux et al. (2018). Varying $\lambda$ smoothly navigates this frontier: increasing $\lambda$ promotes cautious (lower-conflict, higher non-specificity) explanations, whereas decreasing $\lambda$ prioritizes specificity. Figure 4 illustrates how IEMM move along this trade-off as $\lambda$ changes, while IMM remains a static reference, as it does not adapt to the DM's utility function.

Additional experimental details and further tests on synthetic and real-world datasets are provided in Appendix I.

## 4 CONCLUSION

In this paper, we presented a novel approach to explainable evidential clustering using decision trees as explainers. Through the introduction of utility functions, we extended the concept of representativity, a both necessary and sufficient condition for decision trees to function as abductive explainers, to imprecise settings. This allows for the accommodation of "tolerable" mistakes in explanations, making it particularly suitable for evidential contexts. Building on these theoretical foundations, we proposed the evidential mistakeness measure and developed the Iterative Evidential Mistakeness Minimization (IEMM) algorithm. Our approach produces decision trees that effectively explain evidential clustering, advancing the development of both cautious and explainable AI systems.

An important consideration regards the expected audience of the explanations our algorithm creates. Our work implicitly assumes that decision-makers possess domain expertise, an understanding of the implications of their choices, and knowledge about their risk tolerance preferences. The

explanations we generate are designed for these informed stakeholders—individuals familiar with the feature space and its relationships. For example, in clinical applications, our explanations target medical professionals who can appropriately interpret physiological measurements, rather than patients without specialized knowledge.

A notable property of IEMM is the generation of inherently shallow decision trees. Following the IMM design principle, IEMM produces exactly one leaf per cluster, limiting tree depth to at most $|\mathbb{F}| - 1$. This structural constraint enhances interpretability—a primary goal of explainable AI—though it may occasionally result in explanations that cannot fully capture complex data patterns, potentially creating overly rigid explainers for certain applications.

We offer two kinds of guarantees regarding the quality of explanations produced by IEMM. First, when the original evidential clustering is categorical and produced by the Evidential C-Means algorithm, we can bound the explanation cost of the resulting decision tree relative to the original clustering if setting $\mathcal{U}(A, B) = \mathbb{1}_{A=B}$ (see appendix G for details). Second, in the general case with $\mathcal{U}(A, B) \neq \mathbb{1}_{A=B}$, our experiments demonstrate that the IEMM heuristics effectively optimize the corresponding $\mathcal{U}$-evidential representativeness. This produces explanations that align more closely with the cautiousness preferences encoded in the utility function compared to those produced by the $\mathcal{U}(A, B) = \mathbb{1}_{A=B}$ baseline.

Our research inaugurates **perspectives** for future investigation, particularly in two key directions:

- **Elicitation of Utilities in Imprecise Contexts**: While we have proposed a family of natural constructions for utility functions, domain-specific adaptations warrant further exploration in order to better capture the preferences of decision-makers. Future work could focus on developing systematic methods for characterizing and eliciting these utilities.

- **Advanced interpretable evidential classifiers**: Developing more sophisticated interpretable evidential classifiers that exceed the performance of standard decision trees represents a significant opportunity. Potential approaches include incorporating other DT-based explainers into the evidential case Lawless & Gunluk (2022); Fleissner et al. (2024) and constructing Belief Rule-Based Jiao et al. (2015) Explainers, which can incorporate the "non-categoricalness" of the original evidential partition into the explanation. Additionally, extending these methods to better account for the open-world hypothesis could enhance their robustness in real-world applications.

In conclusion, by advancing methods for cautious and explainable clustering, our work contributes to the broader goal of developing AI systems that effectively handle uncertainty while remaining interpretable to human experts. The IEMM algorithm and its theoretical foundations represent a step toward AI systems that acknowledge imperfect information, incorporate domain expertise, and communicate their reasoning in an accessible manner—all key requirements for responsible AI deployment in high-stakes decision-making contexts.

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

# A    EVIDENTIAL THEORY AND CLUSTERING

## A.1    EVIDENTIAL THEORY

From a mass function, one can derive two key set functions—belief and plausibility:

$$\text{Bel}_m(A) = \sum_{B \subseteq A} m(B) \quad \text{and} \quad \text{Pl}_m(A) = \sum_{B \cap A \neq \varnothing} m(B).$$

The belief function $\text{Bel}_m(A)$ for $A \subseteq \Omega$ represents the degree of confidence that 'the correct hypothesis $\omega$ belongs to $A$'. In contrast, the plausibility function $\text{Pl}_m(A)$ captures the degree of confidence that 'it is not impossible for the correct hypothesis $\omega$ to belong to $A$'. Mass functions generalize probability mass functions by distributing belief across all subsets of $\Omega$, rather than only its individual elements.

Another useful measure is the **pignistic probability** Smets (1990), which transforms a mass function into a probability distribution over $\Omega$:

$$\text{BetP}_m(\omega) = \sum_{\{\omega\} \subseteq A \subseteq \Omega} \frac{m(A)}{|A|}.$$

The pignistic probability $\text{BetP}_m$ can be interpreted as providing the best Bayesian approximation to the mass function $m$.

Similarly, there are several techniques to derive the categorical mass closest to a given evidential clustering function Imoussaten (2025). A prominent method relies on the **strong dominance** criterion: for any $\omega, \omega' \in \Omega$, $\omega$ strongly dominates $\omega'$ if and only if $\text{Bel}_m(\{\omega\}) > \text{Pl}_m(\{\omega'\})$. Mapping $m$ to the set of non-strongly dominated clusters yields a categorical mass function.

## A.2    CLUSTERING FUNCTIONS

Figure 5 illustrates various clustering methods for $\Omega = \{\omega_1, \omega_2\}$. While hard clustering assigns each point to exactly one cluster, evidential clustering offers a more sophisticated representation by capturing uncertainty and imprecision. It identifies points that may plausibly belong to multiple clusters (represented in the figure by those primarily associated with $\omega_1 \cup \omega_2$) and accounts for points not clearly associated with any cluster (shown as those predominantly linked to $\varnothing$).

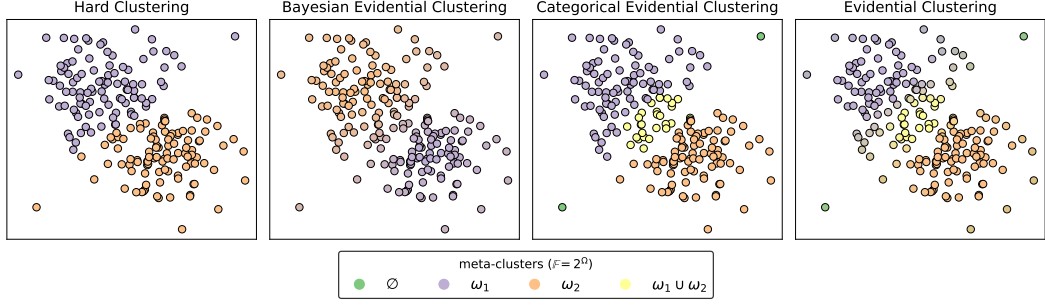

Figure 5: A representation of different clustering functions over a synthetic dataset. In this case, $\mathbb{X} = \mathbb{R}^2$ and $|\Omega| = 2$. Dataset was constructed by sampling 100 points from two normal distributions with centers at $(3, 5)$ and $(5, 3)$ and $\sigma$ of 1. Two outliers were added, at $(2, 2)$ and $(6, 6)$. The evclust package Soubeiga & Antoine (2025) was used to perform clustering. Hard clustering assigns each point to a single cluster. Bayesian evidential clustering gives a membership level for each observation. Categorical evidential clustering introduces the information about in-between points ($\omega_1 \cup \omega_2$) and outliers ($\varnothing$). Finally, evidential clustering combines all the previous information. The gradient of colors for each point visually represents the mass.

# B    ON SIMPLIFICATION EXPLANATION TECHNIQUES

Simplification techniques typically rely on rule extraction methods, encompassing both global and local approaches. Studies have been conducted to assess the quality of these explanations Amgoud &

Ben-Naim (2022). Below, we introduce definitions that characterize effective explanations produced by simplification techniques.

A **feature literal** is a pair $\langle \mathcal{A}, v \rangle$ where $\mathcal{A} \in \mathcal{D}$ and $v \in \mathcal{A}$. Let $\mathbf{L}$ be the set of all feature literals for all attributtes. A consistent subset of feature literals is $L \subset \mathbf{L}$ such that $\langle \mathcal{A}, v \rangle, \langle \mathcal{A}, v' \rangle \in L \Rightarrow v = v'$. Let $\mathbf{C} \subseteq 2^{\mathbf{L}}$ be the set of **all consistent subsets of feature literals**. Each $\mathbf{D} \subset \mathbf{C}$ induces a map $\mathrm{DNF} : \mathbb{X} \to \{\mathrm{True}, \mathrm{False}\}$ with

$$\mathrm{DNF}_{\mathbf{D}}(x) = \bigvee_{L \in \mathbf{D}} \left( \bigwedge_{\langle \mathcal{A}, v \rangle \in L} (x_{\mathcal{A}} = v) \right) \tag{6}$$

which is a Disjunctive Normal Form (DNF) Su et al. (2015). A DNF can serve as a human-interpretable classification model. When a DNF matches the behavior of a black-box classifier, we achieve a particularly desirable outcome known as an abductive explanation.

---

**A Concrete Example:** Consider a philosopher studying living beings who observes two key characteristics: their appearance and their mode of locomotion. To formalize this classification problem, the philosopher defines the feature space of conceivable living beings as $\mathbb{X} = \mathtt{App} \times \mathtt{Move}$, where

$$\mathtt{App} = \{\mathtt{feathered}, \mathtt{featherless}\} \text{ and } \mathtt{Move} = \{\mathtt{biped}, \mathtt{non\text{-}biped}\}.$$

From these features, we can construct the set of feature literals:

$$\mathbf{L} = \{\langle \mathtt{App}, \mathtt{feathered} \rangle, \langle \mathtt{App}, \mathtt{featherless} \rangle, \langle \mathtt{Move}, \mathtt{biped} \rangle, \langle \mathtt{Move}, \mathtt{non\text{-}biped} \rangle\}.$$

The set of all consistent subsets of feature literals encompasses all possible combinations that do not contain contradictory values for the same attribute:

$\mathbf{C} = \{\varnothing,$
$\{\langle \mathtt{App}, \mathtt{feathered} \rangle\}, \{\langle \mathtt{App}, \mathtt{featherless} \rangle\}, \{\langle \mathtt{Move}, \mathtt{biped} \rangle\}, \{\langle \mathtt{Move}, \mathtt{non\text{-}biped} \rangle\},$
$\{\langle \mathtt{App}, \mathtt{feathered} \rangle, \langle \mathtt{Move}, \mathtt{biped} \rangle\}, \{\langle \mathtt{App}, \mathtt{feathered} \rangle, \langle \mathtt{Move}, \mathtt{non\text{-}biped} \rangle\},$
$\{\langle \mathtt{App}, \mathtt{featherless} \rangle, \langle \mathtt{Move}, \mathtt{biped} \rangle\}, \{\langle \mathtt{App}, \mathtt{featherless} \rangle, \langle \mathtt{Move}, \mathtt{non\text{-}biped} \rangle\}$
$\}.$

An example of a DNF is given by $\mathbf{D} = \{\{\langle \mathtt{App}, \mathtt{featherless} \rangle, \langle \mathtt{Move}, \mathtt{biped} \rangle\}\}$, which corresponds to featherless bipedal beings. That is, for any living being $x$, we have $\mathrm{DNF}_{\mathbf{D}}(x) = (x_{\mathtt{App}} = \mathtt{featherless}) \wedge (x_{\mathtt{Move}} = \mathtt{biped})$. $\mathrm{DNF}_{\mathbf{D}}(x)$ is true whenever $x$ is a human being.
Conversely, $\mathbf{D}' = \{\{\langle \mathtt{App}, \mathtt{feathered} \rangle\}, \{\langle \mathtt{Move}, \mathtt{non\text{-}biped} \rangle\}\}$ corresponds to beings that are either feathered or non-bipedal. In this case, the induced DNF is $\mathrm{DNF}_{\mathbf{D}'}(x) = (x_{\mathtt{App}} = \mathtt{feathered}) \vee (x_{\mathtt{Move}} = \mathtt{non\text{-}biped})$ and $\mathrm{DNF}_{\mathbf{D}'}(x)$ is false whenever $x$ is a human being.

---

**Definition 6.** An **abductive explanation** of the label $\omega \in \Omega$ is a $L \in \mathbf{C}$ such that, $\forall x \in \mathbb{X}$,

$$\left( \bigwedge_{\langle \mathcal{A}, v \rangle \in L} (x_{\mathcal{A}} = v) \right) \Rightarrow \Gamma(x) = \omega$$

Abductive explanations were introduced to address the question: "Why is $\Gamma(x) = \omega$?", providing a sufficient reason for characterizing the label $\omega$, where $\Gamma$ is a supervised classifier Ignatiev et al. (2019). In the context of explainability, an ideal construction would be a system that can provide satisfactory explanations[4] for a classifier's outputs.

**Definition 7.** An **explainer** of a classifier $\Gamma : \mathbb{X} \to \Omega$ is a map $\chi_{\Gamma} : \Omega \to 2^{\mathbf{C}}$.

That is, to each class $\omega$, a classifier associates a DNF. If the DNF issued from $\chi_{\Gamma}$ matches $\Gamma$, the classifier provides abductive explanations.

---

[4]In this work, we consider satisfactory explanations to be "abductive" or "as abductive as possible." However, this might not always be the case. As highlighted in multiple works Barredo Arrieta et al. (2020), the best type of explanation depends on the audience for which this explanation is intended. We develop this discussion further in the conclusion.

# C    DECISION TREES AS EXPLAINERS

In this section, we provide a brief overview of decision trees (DTs) and their role as explainers. We also establish the relationship between representativity and abductivity in the context of DTs. We adapt the following definition of univariate decision trees from Izza et al. (2022a).

**Definition 8.** The **graph of a decision tree** $\mathcal{T} = (V, E)$ is a directed acyclic graph in which there is at most one path between any two vertices. The vertex set $V$ is divided into non-terminal vertices $N$ and terminal vertices $T$, such that $V = N \cup T$. Additionally, $\mathcal{T}$ has a unique root vertex, $\text{root}(\mathcal{T}) \in V$, which has no incoming edges, while every other vertex has exactly one incoming edge.

To each graph of a DT, there are two important associated functions:

- A **split** is a map $\phi : N \to \mathcal{D}$ that assigns an attribute to each non-terminal vertex.

- Let $\text{children}(r) = \{s \in V \mid (r, s) \in E\}$ be the set of children of a vertex $r$. A **decision** is a map $\varepsilon : E \to \mathbf{L}$ such that, for every non-terminal vertex $r \in N$, there exists a bijection $\varepsilon_r : \text{children}(r) \to \phi(r)$ satisfying $\varepsilon(r, s) = \langle \phi(r), \varepsilon_r(s) \rangle$.

It is well known that any binary decision tree can be transformed in linear time into an equivalent disjunctive normal form (DNF) expression Audemard et al. (2022). This property is often referenced when DTs are described as "interpretable" Guidotti et al. (2018). With this in mind, we associate each vertex with a path, which serves as the foundation for interpreting a decision tree as an explainer.

For a fixed graph of a DT $\mathcal{T}$, let $\text{DNF}(r)$ be the set of literals associated with the edges that link the root to vertex $r$. All literals in $\text{DNF}(r)$ are consistent Izza et al. (2022a). That is, $\text{DNF} : V \to 2^{\mathbf{C}}$. Let $\mathbf{D} = \text{DNF}(T)$ be the set of all DNFs associated with terminal vertices.

**Definition 9.** A **path** is a map $\Upsilon : \mathbb{X} \to \mathbf{D}$ such that, for all $x \in \mathbb{X}$,

$$\bigwedge_{\langle \mathcal{A}, v \rangle \in \Upsilon(x)} (x_{\mathcal{A}} = v).$$

The partitioning nature of decision trees ensures that each path is well-defined, meaning every possible observation follows a unique path. This characteristic allows us to interpret vertices as subsets of the feature space Hoarau et al. (2023a).

**Definition 10.** A **node** is a nonempty subset $S \subseteq \mathbb{X}$.

Every achievable vertex can be trivially associated with a unique node by its DNF. We call **leaves** the nodes associated with terminal vertices. The set $\mathbf{D}$ can be understood as the explanation for each leaf. Associating leaves with explanations allows us to define the DT as a classifier.

**Definition 11.** A **decision tree** is a map $\Delta : \mathbb{X} \to \Omega$ to which a path $\Upsilon_{\Delta}$ provides an abductive explanation. That is, $\forall x \in \mathbb{X}$,

$$\bigwedge_{\langle \mathcal{A}, v \rangle \in \Upsilon(x)} (x_{\mathcal{A}} = v) \Rightarrow \Delta(x) = \omega.$$

Let, $\forall \omega \in \Omega$, $\mathcal{L}_{\omega}^{\Delta} = \{\Upsilon^{-1}(\{L\}) : L \in \Upsilon(\Delta^{-1}(\{\omega\}))\}$ be the set of all leaves associated with the explanation of $\omega$. The **DT explainer** $\chi_{\Gamma}^{\Delta}$ associated with $\Delta$ is an explainer that, for any label, returns all paths explaining it. That is, $\chi_{\Gamma}^{\Delta}(\omega) = \Upsilon_{\Delta}[\mathcal{L}_{\omega}^{\Delta}] = \{\Upsilon_{\Delta}(x) : x \in \mathcal{L}_{\omega}^{\Delta}\}$.

Our investigation focuses on the quality of explanations when, in the context of model simplification, the original classifier diverges from the decision tree explaining it. We borrow the concept of representative explainer from Amgoud & Ben-Naim (2022). A representative explainer is one that, for all observations $x$ with label $\omega = \Gamma(x)$, can provide an explanation $L$ that holds at $x$. That is, there exists a set of literals $L \in \chi_{\Gamma}(\omega)$ such that $\langle \mathcal{A}, v \rangle \in L \Rightarrow x_{\mathcal{A}} = v$.

**Definition 12.** A **representative explainer** is an explainer $\chi_{\Gamma}$ such that, $\forall \omega \in \Omega$, $\forall x \in \Gamma^{-1}(\{\omega\})$, $\exists L \in \chi_{\Gamma}(\omega)$ such that, $\forall \langle \mathcal{A}, v \rangle \in L, x_{\mathcal{A}} = v$.

The work in Amgoud & Ben-Naim (2022) proves that every explainer providing abductive explanations is representative. We complement this result by proving that every representative DT explainer provides abductive explanations. Thus, for DT explainers, representativity and abductivity are equivalent.

**Theorem 1.** *If the $\chi_\Gamma^\Delta$ explainer is representative, it provides abductive explanations.*

*Proof.* We proceed by contradiction. Assume the DT explainer does not provide abductive explanations.

From definition 11, this implies that $\Gamma \neq \Delta$. That is, there exists $x \in \mathbb{X}$ such that $\omega_\Gamma = \Gamma(x) \neq \Delta(x) = \omega_\Delta$. Since the leaves form a partition of the feature space and $x \in \mathcal{L}_{\omega_\Delta}^\Delta$, we have $\Upsilon_\Delta(x) \notin \Upsilon_\Delta[\mathcal{L}_{\omega_\Gamma}^\Delta]$, and the explainer is not representative. $\square$

## D  ON EXPLANATION COSTS

In the context of a categorical evidential partition $\overline{\mathcal{M}}_c$, we want to characterize the cost of explaining a point $x$ with a cautious explainer induced by some interpretable classifier $\Delta : X \to 2^\Omega$.

The utility $\mathcal{U}(A, \overline{\mathcal{M}}_c(x))$ quantifies the satisfaction of assigning metacluster $A$ to observation $x$ and, therefore, equals the cost of not assigning $A$ to $x$.

Thus, the total cost can be understood as the function $\overline{\text{Cost}_{\mathcal{M}_c, \Delta}} : X \to [0, |\mathbb{F}| - 1]$, that maps $x$ to the sum of costs from not assigning $x$ to all metaclusters $A \neq \Delta(x)$,

$$\overline{\text{Cost}_{\mathcal{M}_c, \Delta}}(x) = \sum_{A \neq \Delta(x)} \mathcal{U}(A, \overline{\mathcal{M}}_c(x)). \tag{7}$$

Conversely, $1 - \mathcal{U}(A, \overline{\mathcal{M}}_c(x))$ represents the cost of assigning $A$ to $x$, leading to $\underline{\text{Cost}_{\mathcal{M}_c, \Delta}} : X \to [0, 1]$, an alternative expression for total cost:

$$\underline{\text{Cost}_{\mathcal{M}_c, \Delta}}(x) = 1 - \mathcal{U}(\Delta(x), \overline{\mathcal{M}}_c(x)). \tag{8}$$

It always holds that $\underline{\text{Cost}_{\mathcal{M}_c, \Delta}}(x) \leq \overline{\text{Cost}_{\mathcal{M}_c, \Delta}}(x)$. When $\mathcal{U}(A, B) = \mathbb{1}_{A = B}$, the equality holds. Furthermore, if $\mathcal{M}_c$ induces a hard clustering and $\Delta$ is IMM-like (with exactly one centroid per leaf), both equal one (and not zero) if and only if $x$ is a mistake as stated in Definition 2.

### D.1  A SIMPLE EXAMPLE

Figure 6 and Table 1 illustrate the utility concept through a concrete example. The value $\mathcal{U}(\{\omega_1, \omega_2\}, \{\omega_1\})$ quantifies how tolerable the mistake at $x_1$ is. When $\mathcal{U}(\{\omega_1, \omega_2\}, \{\omega_1\}) = 0$, this mistake becomes as intolerable as the one at $x_2$. Conversely, when $\mathcal{U}(\{\omega_1, \omega_2\}, \{\omega_1\}) = 1$, it becomes as tolerable as the correct assignment at $x_0$. In this latter scenario, removing $x_2$ from the dataset would yield an optimal assignment.

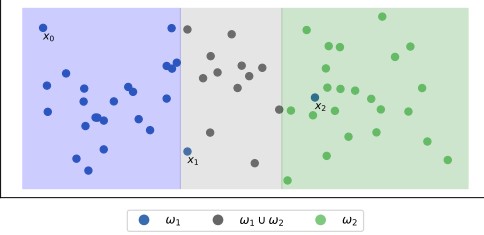

| $x$ | $x_0$ | $x_1$ | $x_2$ |
|---|---|---|---|
| $\Delta(x)$ | $\{\omega_1\}$ | $\{\omega_1, \omega_2\}$ | $\{\omega_2\}$ |
| $\underline{\text{Cost}_{\mathcal{M}_c, \Delta}}(x)$ | 0 | $1 - \mathcal{U}(\{\omega_1, \omega_2\}, \{\omega_1\})$ | 1 |
| $2^\Omega \setminus \Delta(x)$ | $\{\{\omega_2\}, \{\omega_1, \omega_2\}\}$ | $\{\{\omega_1\}, \{\omega_2\}\}$ | $\{\{\omega_1\}, \{\omega_1, \omega_2\}\}$ |
| $\overline{\text{Cost}_{\mathcal{M}_c, \Delta}}(x)$ | $\mathcal{U}(\{\omega_1, \omega_2\}, \{\omega_1\})$ | 1 | $1 + \mathcal{U}(\{\omega_1, \omega_2\}, \{\omega_1\})$ |

Table 1: For each point highlighted in Figure 6, we present the point $x$, the metacluster assigned by classifier $\Delta$, the cost of assigning $x$ to the metacluster designated by $\Delta$, the set of metaclusters not assigned by $\Delta$, and the cost of not assigning $x$ to them. Note that $\overline{\mathcal{M}}_c(x_0) = \overline{\mathcal{M}}_c(x_1) = \overline{\mathcal{M}}_c(x_2) = \{\omega_1\}$.

Figure 6: Illustration of a categorical evidential classifier and space partition in $\mathbb{X} = \mathbb{R}^2$. The partition $\Delta$ separates $x_1$ and $x_2$ from their respective metaclusters, while correctly assigning all other observations.

# E  RELATING MISTAKENESS AND REPRESENTATIVENESS

In this section, we show that the evidential representativeness and the total evidential mistakeness (the sum of the evidential mistakeness of each leaf) are equivalent in terms of measuring the quality of a IMM-like decision tree.

**Theorem 2.** *Let $\Delta, \Delta' : \mathbb{X} \to 2^\Omega$ be two IMM-like decision trees. Then, for any evidential partition $\mathcal{M}$ and utility $\mathcal{U}$,*

$$\mathcal{R}_{\mathcal{M},\mathcal{U}}(\Delta) \geq \mathcal{R}_{\mathcal{M},\mathcal{U}}(\Delta')$$

$$\iff \sum_{A \subset \Omega} \overline{M}_{\mathcal{M},\mathcal{U}}(\mathcal{L}_A^\Delta) \leq \sum_{A \subset \Omega} \overline{M}_{\mathcal{M},\mathcal{U}}(\mathcal{L}_A^{\Delta'})$$

$$\iff \sum_{A \subset \Omega} \underline{M}_{\mathcal{M},\mathcal{U}}(\mathcal{L}_A^\Delta) \leq \sum_{A \subset \Omega} \underline{M}_{\mathcal{M},\mathcal{U}}(\mathcal{L}_A^{\Delta'})$$

*where $v_A \in \mathcal{L}_A^\Delta$ which is the leaf associated with the cluster $A$ in the decision tree $\Delta$.*

*Proof.* We start by establishing the relation between the two mistakenness functions. By definition, $x \in \mathcal{L}_A^\Delta \iff \Delta(x) = A$. From equations (4) and (5),

$$\overline{M}_{\mathcal{M},\mathcal{U}}(\mathcal{L}_A^\Delta) = \sum_{x \in \mathcal{L}_A^\Delta} \sum_{\Delta(x) \neq C} \sum_{B \in \mathbb{F}_\mathcal{M}} \mathcal{U}(C,B) m_x(B),$$

$$\underline{M}_{\mathcal{M},\mathcal{U}}(\mathcal{L}_A^\Delta) = \sum_{x \in \mathcal{L}_A^\Delta} \sum_{B \in \mathbb{F}_\mathcal{M}} (1 - \mathcal{U}(A,B)) m_x(B).$$

Then,

$$\sum_{A \subset \Omega} \underline{M}_{\mathcal{M},\mathcal{U}}(\mathcal{L}_A^\Delta) - \sum_{A \subset \Omega} \overline{M}_{\mathcal{M},\mathcal{U}}(\mathcal{L}_A^\Delta)$$

$$= \sum_{A \subset \Omega} \sum_{x \in \mathcal{L}_A^\Delta} \sum_{B \in \mathbb{F}_\mathcal{M}} m_x(B) \left( (1 - \mathcal{U}(A,B)) - \sum_{A \neq C} \mathcal{U}(C,B) \right)$$

$$= \sum_{x \in X} \sum_{B \in \mathbb{F}_\mathcal{M}} m_x(B) \left( 1 - \sum_{C \subset \Omega} \mathcal{U}(C,B) \right)$$

$$= |X| - \sum_{x \in X} \sum_{B \in \mathbb{F}_\mathcal{M}} m_x(B) \left( \sum_{C \subset \Omega} \mathcal{U}(C,B) \right) = |X| - \kappa_{\mathcal{M},\mathcal{U}}.$$

where $\kappa_{\mathcal{M},\mathcal{U}}$ is a constant that depends only on the evidential partition $\mathcal{M}$ and utility $\mathcal{U}$, but not on the specific decision tree $\Delta$.

Also, from equation (3),

$$|X| \mathcal{R}_{\mathcal{M},\mathcal{U}}(\Delta) = \sum_{A \subset \Omega} \sum_{x \in \mathcal{L}_A^\Delta} \sum_{B \in \mathbb{F}_\mathcal{M}} \mathcal{U}(\Delta(x), B) m_x(B).$$

Similarly,

$$|X| \mathcal{R}_{\mathcal{M},\mathcal{U}}(\Delta) + \sum_{A \subset \Omega} \overline{M}_{\mathcal{M},\mathcal{U}}(\mathcal{L}_A^\Delta) =$$

$$\sum_{A \subset \Omega} \sum_{x \in \mathcal{L}_A^\Delta} \sum_{B \in \mathbb{F}_\mathcal{M}} m_x(B) \left( \mathcal{U}(A,B) + \sum_{A \neq C} \mathcal{U}(C,B) \right) = \kappa_{\mathcal{M},\mathcal{U}}.$$

Since all three measures are related by affine transformations with the same constant terms, they preserve the same ordering relationships between different decision trees. Therefore, comparing two trees $\Delta$ and $\Delta'$ using any of these measures yields equivalent results, proving the stated equivalences. $\qquad\square$

# F  CHOOSING A UTILITY FUNCTION

When an explainer yields a metacluster $A$, while the original classifier assigns $B$, two types of errors can occur. The first is insufficient coverage, measured by $|A^C \cap B|$ - where the explainer fails to include all elements of the true metacluster. The second is excessive coverage, measured by $|A \cap B^C|$ - where the explainer includes elements not in the true metacluster. Penalizing insufficient coverage indicates the explainer is not cautious enough, while penalizing excessive coverage suggests it is too cautious.

To address both error types, we introduce two families of utility functions with a positive parameter $\lambda$ controlling their behavior:

$$\overline{\mathcal{U}}^\lambda(A, B) = \left( \frac{|A \cap B|}{|A \cup B|} \mathbb{1}_{B \subset A} \right)^{1/\lambda} \text{ and } \underline{\mathcal{U}}^\lambda(A, B) = \left( \frac{|A \cap B|}{|A \cup B|} \mathbb{1}_{A \subset B} \right)^{1/\lambda}.$$

These utility functions exhibit complementary tolerance behaviors. The function $\overline{\mathcal{U}}^\lambda(A, B)$ assigns zero utility when $A^C \cap B \neq \varnothing$, making it completely intolerant to insufficient coverage while allowing parameter $\lambda$ to modulate tolerance to excessive coverage. Higher $\lambda$ values reduce penalties for excessive coverage, embodying a more cautious approach. In contrast, $\underline{\mathcal{U}}^\lambda(A, B)$ assigns zero utility when $A \cap B^C \neq \varnothing$, showing complete intolerance to excessive coverage while $\lambda$ controls the degree of tolerance to insufficient coverage—higher $\lambda$ values reducing penalties for insufficient coverage and representing a more risk-attractive approach.

These combine into a comprehensive family of utility functions for $\lambda \in \mathbb{R}^*$:

$$\mathcal{U}^\lambda(A, B) = \begin{cases} \underline{\mathcal{U}}^{|\lambda|}(A, B) & \text{if } \lambda < 0 \\ \overline{\mathcal{U}}^{|\lambda|}(A, B) & \text{if } \lambda > 0 \end{cases}.$$

We also define special cases as limits when $\lambda$ approaches 0 and $\pm\infty$. That gives $\mathcal{U}^0(A, B) = \mathbb{1}_{A=B}$, $\mathcal{U}^{-\infty}(A, B) = \mathbb{1}_{A \subset B}$ and $\mathcal{U}^\infty(A, B) = \mathbb{1}_{B \subset A}$.

Finally, based on these, we define the $\lambda$-**evidential mistakeness** as:

$$M_\mathcal{M}^\lambda = \begin{cases} \overline{M}_{\mathcal{M}, \mathcal{U}^\lambda} & \text{if } \lambda \geq 0 \\ \underline{M}_{\mathcal{M}, \mathcal{U}^\lambda} & \text{if } \lambda < 0 \end{cases} \tag{9}$$

for any $\lambda \in \mathbb{R} \cup \{\pm\infty\}$. The higher the $\lambda$, the more the mistakeness function represents a conservative approach. When the underlying clustering function is hard and $\lambda = 0$ is chosen, the algorithm 1 operates identically to IMM because evidential mistakeness equals the number of mistakes. For cautious partitions as input, varying $\lambda$ controls the "level of cautiousness" of the resulting explainer. Figure 7 illustrates how different $\lambda$ values influence the partitioning of the feature space by IEMM.

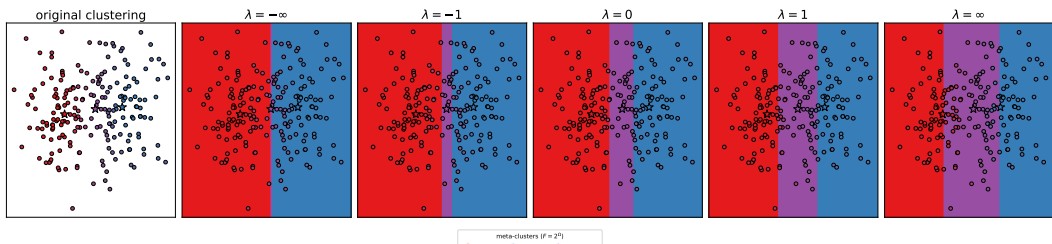

Figure 7: An example, based on a given two-features clustering (column 1), of IEMM partitioning the feature space. The stars represent the centroid of each metacluster. The utility function strongly influences the resulting explanations. The higher the $\lambda$, the more the explainer assigns larger portions of the space to metaclusters representing doubt. At the limit, $\lambda = -\infty$ (column 2), the obtained explanations give the maximum possible space to the singleton metaclusters. Conversely, $\lambda = \infty$ (column 6) assigns the maximum possible space to the metaclusters representing doubt.

# G  BOUNDING THE EXPLANATION COST

Let $\overline{\mathcal{M}}_c : X \to 2^\Omega \setminus \{\varnothing\}$ be a categorical evidential partition. Its evidential $c$-means cost Masson & Denœux (2008) is

$$\mathcal{J}_{ECM}(\overline{\mathcal{M}}_c) = \sum_{x \in X} |\overline{\mathcal{M}}_c(x)|^\alpha \left\| x - v_{\overline{\mathcal{M}}_c(x)} \right\|_2^2.$$

where $\alpha \geq 0$ controls the penalty for imprecision (larger focal sets) and $v = \{v_A \in X : A \in \mathbb{F}_{\overline{\mathcal{M}}_c}\}$ is the set of metacluster centroids.

For an interpretable classifier $\Delta : X \to 2^\Omega$, its cost is

$$\mathcal{J}_{ECM}(\Delta) = \sum_{x \in X} |\Delta(x)|^\alpha \left\| x - v_{\Delta(x)} \right\|_2^2.$$

We consider the case where IEMM (Algorithm 1) is run with utility $\mathcal{U}(A, B) = \mathbb{1}_{A=B}$ to construct a decision tree explainer $\Delta$ of the partition $\overline{\mathcal{M}}_c$. Under this utility, IEMM behaves as IMM over the metaclusters. Hence we can transfer the structural bounds from Moshkovitz et al. (2020) to the categorical evidential setting and relate $\mathcal{J}_{ECM}(\Delta)$ to $\mathcal{J}_{ECM}(\overline{\mathcal{M}}_c)$.

**Theorem 3.** *If IEMM takes $(X, \overline{\mathcal{M}}_c, \mathbb{F}_{\overline{\mathcal{M}}_c}, v)$ as input, uses $\mathcal{U}(A, B) = \mathbb{1}_{A=B}$ to define mistakeness, and outputs a decision tree $\Delta$ of height $H$, then*

$$\mathcal{J}_{ECM}(\Delta) \leq |\Omega|^\alpha (2 + 8H|\mathbb{F}_{\overline{\mathcal{M}}_c}|) \cdot \mathcal{J}_{ECM}(\overline{\mathcal{M}}_c).$$

*Proof.* Let $\mathrm{Nodes}(\Delta)$ denote the set of nodes in $\Delta$, and for each node $S \in \mathrm{Nodes}(\Delta)$ let $t_S$ be the number of misclassified points produced by the split at $S$. We adapt the proof technique of Moshkovitz et al. (2020) to the categorical evidential clustering setting. The argument hinges on the following two lemmas:

**Lemma 4.**

$$\mathcal{J}_{ECM}(\Delta) \leq 2|\Omega|^\alpha \cdot \sum_{x \in X} \left\| x - v_{\overline{\mathcal{M}}_c(x)} \right\|_2^2 + 2|\Omega|^\alpha \cdot \sum_{S \in \mathrm{Nodes}(\Delta)} t_S \max_{v_A, v_B \in S} \| v_A - v_B \|_2^2$$

**Lemma 5.**

$$\sum_{S \in \mathrm{Nodes}(\Delta)} t_S \max_{v_A, v_B \in S} \| v_A - v_B \|_2^2 \leq 4H |\mathbb{F}_{\overline{\mathcal{M}}_c}| \cdot \sum_{x \in X} \left\| x - v_{\overline{\mathcal{M}}_c(x)} \right\|_2^2.$$

We prove Lemma 4 below. Lemma 5 was proved at Moshkovitz et al. (2020).

Combining Lemmas 4 and 5, we obtain:

$$\mathcal{J}_{ECM}(\Delta) \leq 2|\Omega|^\alpha \cdot \sum_{x \in X} \left\| x - v_{\overline{\mathcal{M}}_c(x)} \right\|_2^2 + 2|\Omega|^\alpha \cdot 4H|\mathbb{F}_{\overline{\mathcal{M}}_c}| \cdot \sum_{x \in X} \left\| x - v_{\overline{\mathcal{M}}_c(x)} \right\|_2^2$$

$$= |\Omega|^\alpha (2 + 8H|\mathbb{F}_{\overline{\mathcal{M}}_c}|) \cdot \sum_{x \in X} \left\| x - v_{\overline{\mathcal{M}}_c(x)} \right\|_2^2.$$

Since $|\overline{\mathcal{M}}_c(x)| \geq 1$ for all $x \in X$, we conclude

$$\mathcal{J}_{ECM}(\Delta) \leq |\Omega|^\alpha (2 + 8H|\mathbb{F}_{\overline{\mathcal{M}}_c}|) \cdot \mathcal{J}_{ECM}(\overline{\mathcal{M}}_c).$$

$\square$

## G.1  PROOF OF LEMMA 4

*Proof.* We adapt the proof technique from Moshkovitz et al. (2020) to the evidential setting.

Partition $X$ into correctly classified points $X^{\text{cor}} = \{x : \Delta(x) = \overline{\mathcal{M}}_c(x)\}$ and misclassified points $X^{\text{mis}} = \{x : \Delta(x) \neq \overline{\mathcal{M}}_c(x)\}$. Then

$$\mathcal{J}_{ECM}(\Delta) = \sum_{x \in X^{\text{cor}}} |\Delta(x)|^\alpha \left\| x - v_{\Delta(x)} \right\|_2^2 + \sum_{x \in X^{\text{mis}}} |\Delta(x)|^\alpha \left\| x - v_{\Delta(x)} \right\|_2^2$$

$$= \sum_{x \in X^{\text{cor}}} \left| \overline{\mathcal{M}}_c(x) \right|^\alpha \left\| x - v_{\overline{\mathcal{M}}_c(x)} \right\|_2^2 + \sum_{x \in X^{\text{mis}}} |\Delta(x)|^\alpha \left\| x - v_{\Delta(x)} \right\|_2^2.$$

By the Cauchy–Schwarz inequality, $(a+b)^2 \leq 2(a^2 + b^2)$. Applying to $\|x - v_{\Delta(x)}\|_2$ with decomposition $x - v_{\Delta(x)} = (x - v_{\overline{\mathcal{M}}_c(x)}) + (v_{\overline{\mathcal{M}}_c(x)} - v_{\Delta(x)})$, we obtain

$$\mathcal{J}_{ECM}(\Delta) \leq \sum_{x \in X^{\text{cor}}} \left| \overline{\mathcal{M}}_c(x) \right|^\alpha \left\| x - v_{\overline{\mathcal{M}}_c(x)} \right\|_2^2$$

$$+ 2 \sum_{x \in X^{\text{mis}}} |\Delta(x)|^\alpha \left( \left\| x - v_{\overline{\mathcal{M}}_c(x)} \right\|_2^2 + \left\| v_{\overline{\mathcal{M}}_c(x)} - v_{\Delta(x)} \right\|_2^2 \right).$$

Since $|\Delta(x)| \leq |\Omega|$ and $|\overline{\mathcal{M}}_c(x)| \leq |\Omega|$, it follows that

$$\mathcal{J}_{ECM}(\Delta) \leq 2|\Omega|^\alpha \sum_{x \in X^{\text{cor}}} \left\| x - v_{\overline{\mathcal{M}}_c(x)} \right\|_2^2$$

$$+ 2|\Omega|^\alpha \sum_{x \in X^{\text{mis}}} \left( \left\| x - v_{\overline{\mathcal{M}}_c(x)} \right\|_2^2 + \left\| v_{\overline{\mathcal{M}}_c(x)} - v_{\Delta(x)} \right\|_2^2 \right).$$

$$= 2|\Omega|^\alpha \sum_{x \in X} \left\| x - v_{\overline{\mathcal{M}}_c(x)} \right\|_2^2 + 2|\Omega|^\alpha \sum_{x \in X^{\text{mis}}} \left\| v_{\overline{\mathcal{M}}_c(x)} - v_{\Delta(x)} \right\|_2^2$$

We now bound the contribution of misclassifications. Associate to each misclassified $x$ the first node $S$ at which the split separates $v_{\overline{\mathcal{M}}_c(x)}$ and $v_{\Delta(x)}$. Until $S$, both centroids remain in the candidate set, hence both focal elements belong to $S$. Therefore

$$\left\| v_{\overline{\mathcal{M}}_c(x)} - v_{\Delta(x)} \right\|_2^2 \leq \max_{v_A, v_B \in S} \| v_A - v_B \|_2^2.$$

Let $t_S$ be the number of misclassified points first separated at node $S$. Summing over nodes yields

$$\sum_{x \in X^{\text{mis}}} \left\| v_{\overline{\mathcal{M}}_c(x)} - v_{\Delta(x)} \right\|_2^2 \leq \sum_{S \in \text{Nodes}(\Delta)} t_S \max_{v_A, v_B \in S} \| v_A - v_B \|_2^2.$$

$\square$

# H    ILLUSTRATIVE EXAMPLE

In this illustrative example, we consider a medical use case where a clinician seeks simple rules to characterize patient profiles into two clusters based on routine body-composition measurements. We use 12 measurements from 251 patients in the Body Fat Prediction Dataset from Kaggle[5]. We first run Evidential c-Means (ECM) Masson & Denœux (2008) to obtain an evidential clustering $\mathcal{M}$, and then apply our IEMM procedure to learn a decision-tree explainer that approximates $\mathcal{M}$ while reflecting the decision-maker's preferred level of cautiousness.

Figure 8 summarizes the marginal distributions of the 12 measurements and how ECM organizes them into metaclusters from which we derive explanations.

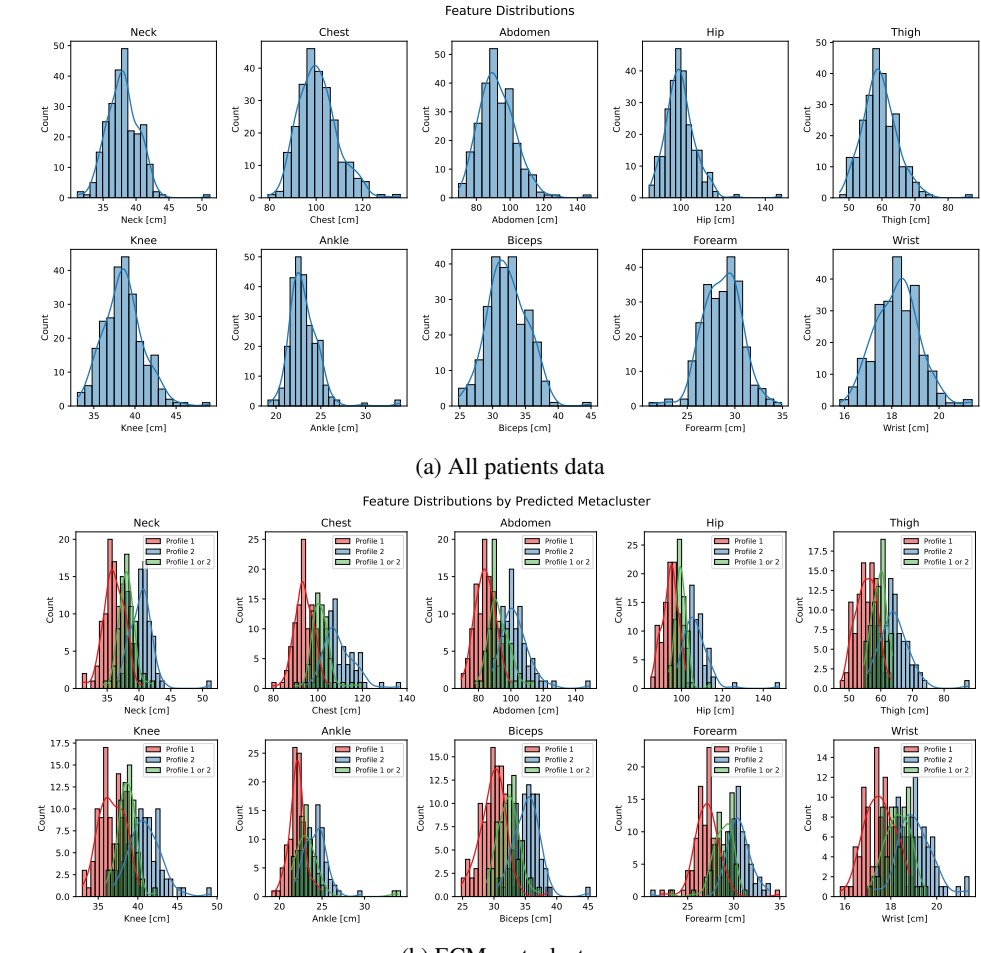

(a) All patients data

(b) ECM metaclusters

Figure 8: Body Fat dataset: distributions of the 12 measurements (top) and the same distributions colored by ECM metaclusters (bottom).

Next, to obtain the actual rules to apply, we learn explainable evidential clusters using IEMM (Algorithm 1), setting the $\lambda$-evidential mistakeness $M_{\mathcal{M}}^{\lambda}$ with $\lambda \in \{-1, 0, 1\}$. For each $\lambda$, Figures 11–10 show the learned decision tree (top subfigure) and the induced distribution of measurements over the resulting evidential assignments (bottom subfigure). The obtained rules vary for different values of $\lambda$, allowing the incorporation of the decision-maker's strategy between cautiousness and specificity into the resulting rule. Across all settings, the induced histograms remain clinically coherent with the ECM structure while offering simple, auditable rules that practitioners can adopt for front-door characterization of patient types.

---

[5] https://www.kaggle.com/datasets/fedesoriano/body-fat-prediction-dataset/data

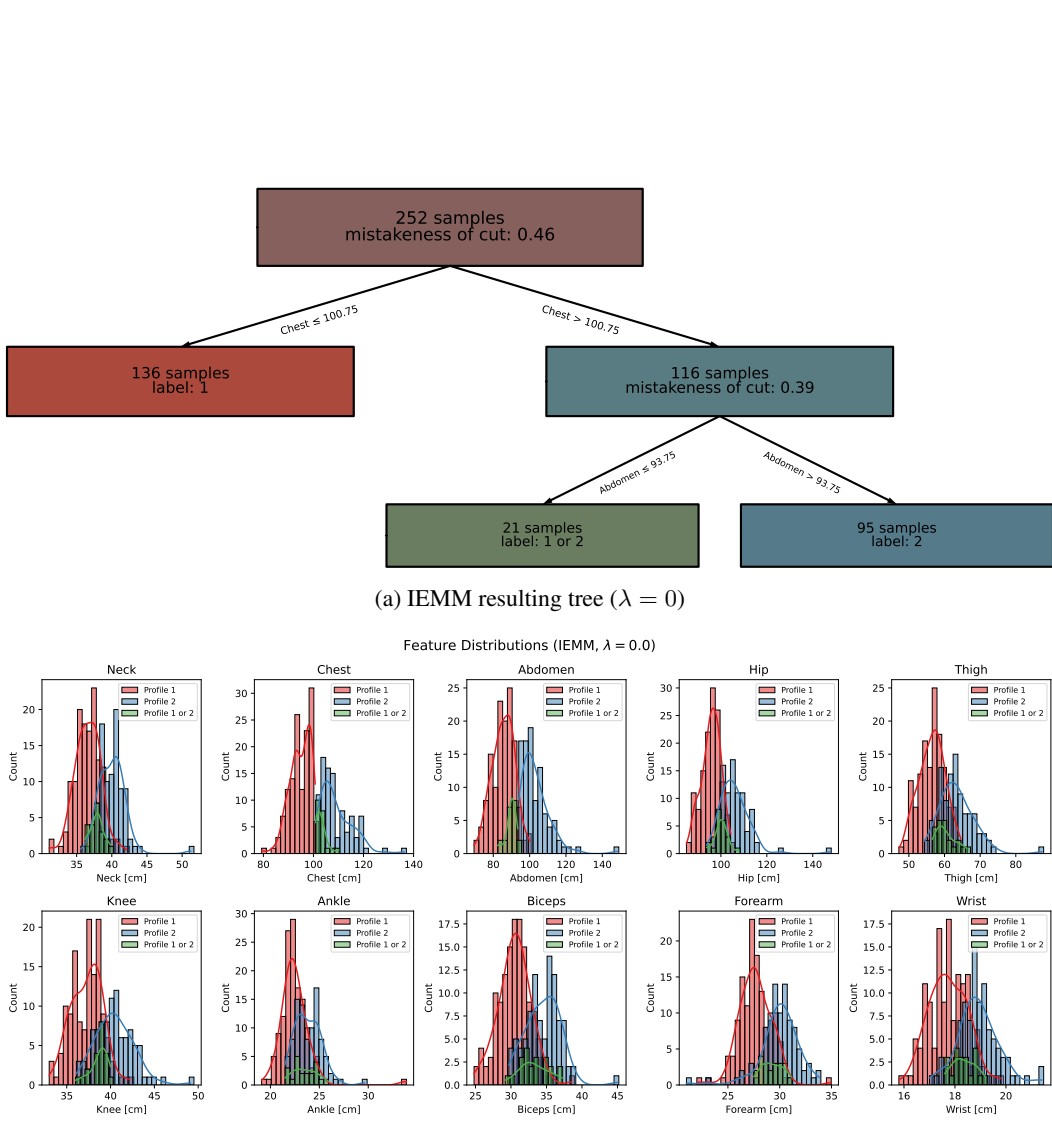

(a) IEMM resulting tree ($\lambda = 0$)

(b) Assignments histograms ($\lambda = 0$)

Figure 9: IEMM with $\lambda = 0$ (uniform weighting): a balanced explainer aligned with ECM meta-clusters without favoring cautiousness or specificity. Measures of the Chest girth and Abdomen girth are key discriminators.

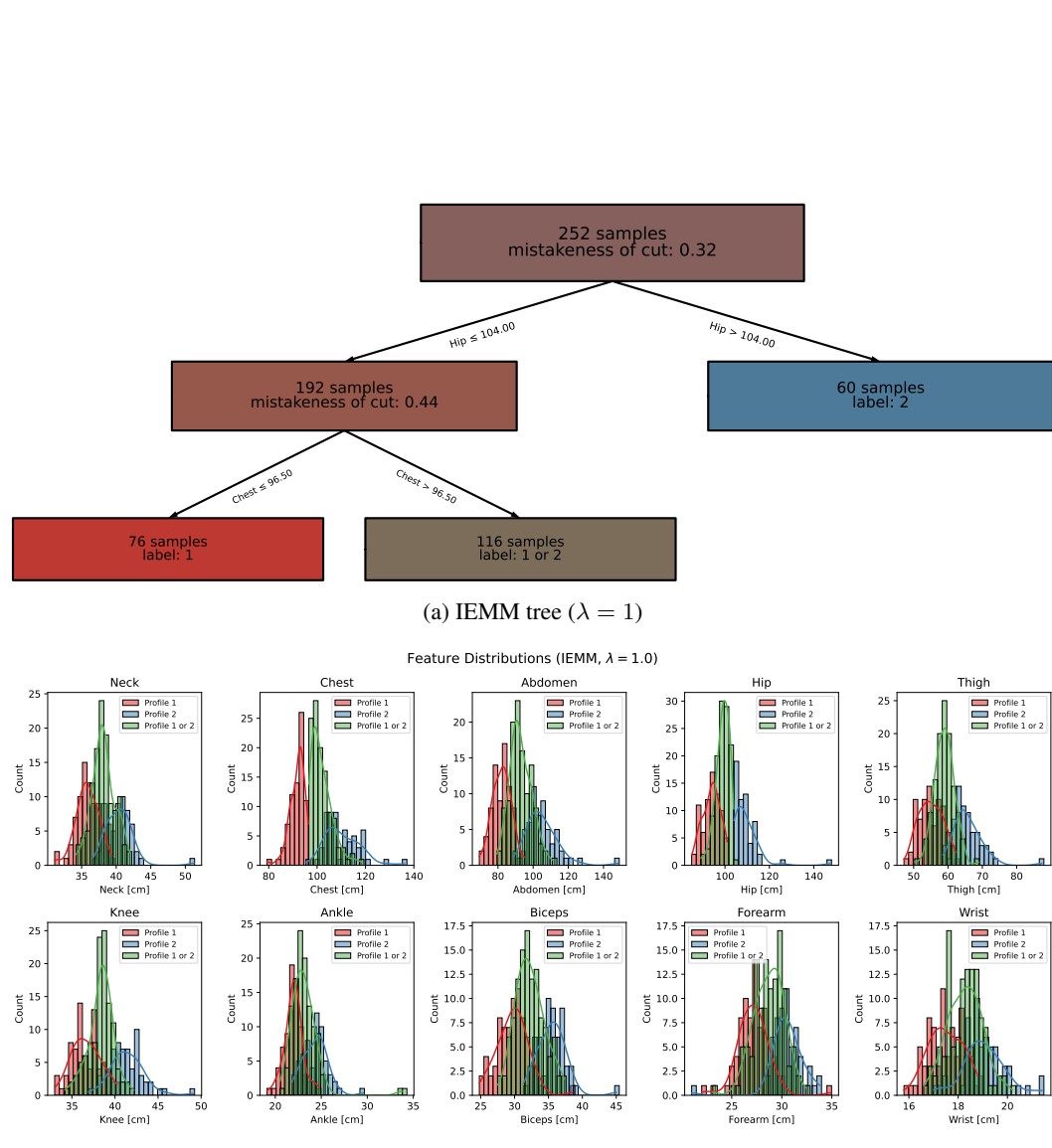

(a) IEMM tree ($\lambda = 1$)

(b) Assignments histograms ($\lambda = 1$)

Figure 10: IEMM with $\lambda = 1$ (cautious): the explainer favors less conflicting and more non-specific metaclusters for ambiguous patients, resulting in more patients (116) being assigned to the imprecise metacluster $\{1, 2\}$. Measures of the Hip girth and Chest girth are key discriminators.

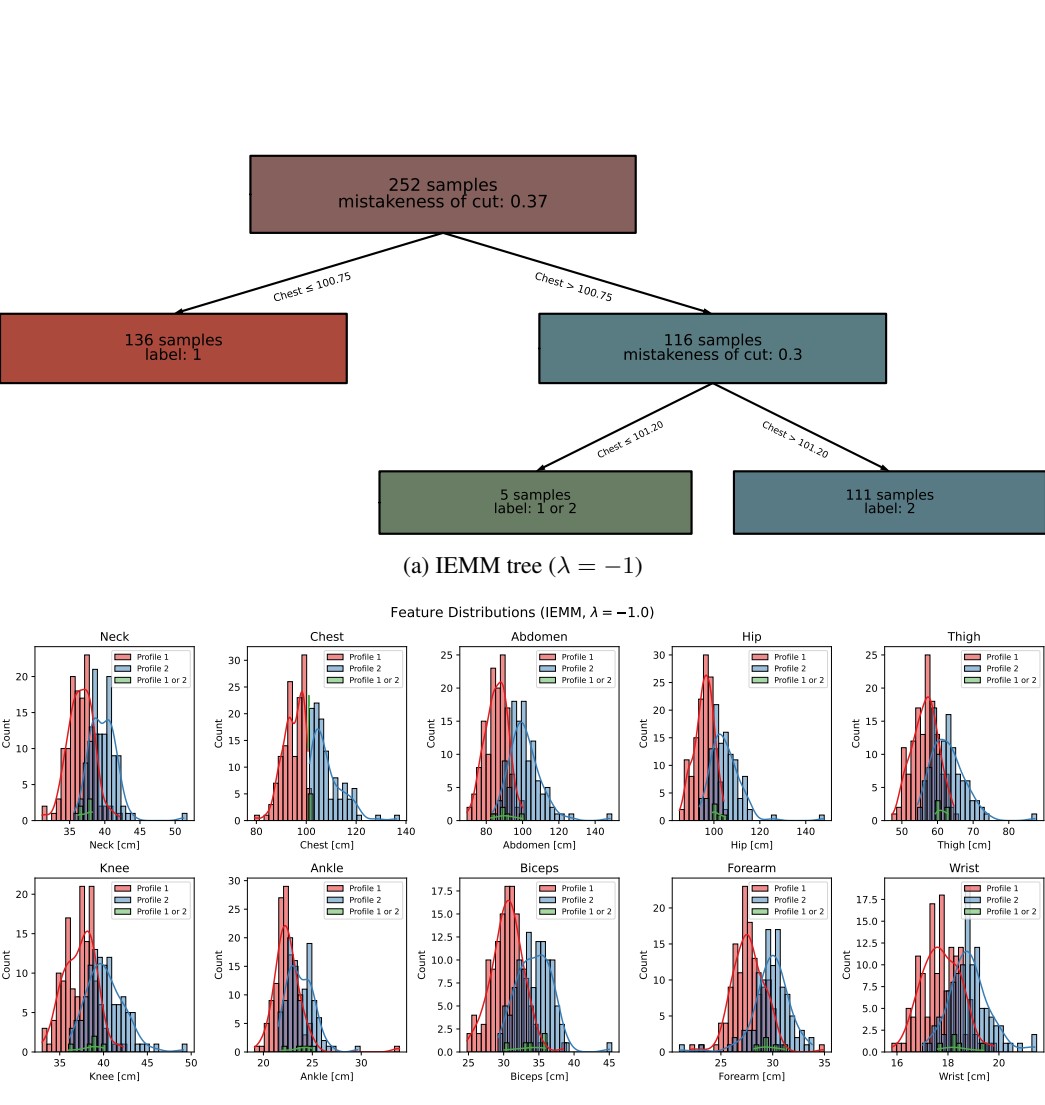

(a) IEMM tree ($\lambda = -1$)

(b) Assignments histograms ($\lambda = -1$)

Figure 11: IEMM with $\lambda = -1$ (specificity-seeking): the explainer prioritizes specific metaclusters, possibly increasing conflict for borderline patients, resulting in fewer patients (5) being assigned to the imprecise metacluster $\{1, 2\}$. The measure of the Chest girth is a key discriminator.

# I  EXPERIMENTS

This section presents additional experimental results that validate the IEMM algorithm. We have implemented IEMM using Python 3.13.6. All code is available at OMMITED TO AVOID IDEN-TIFICATION. The implementation of a decision tree accepting evidential labels was based on the code made available by Hoarau et al. (2023a).

## I.1  TESTS ON SYNTHETIC DATASETS

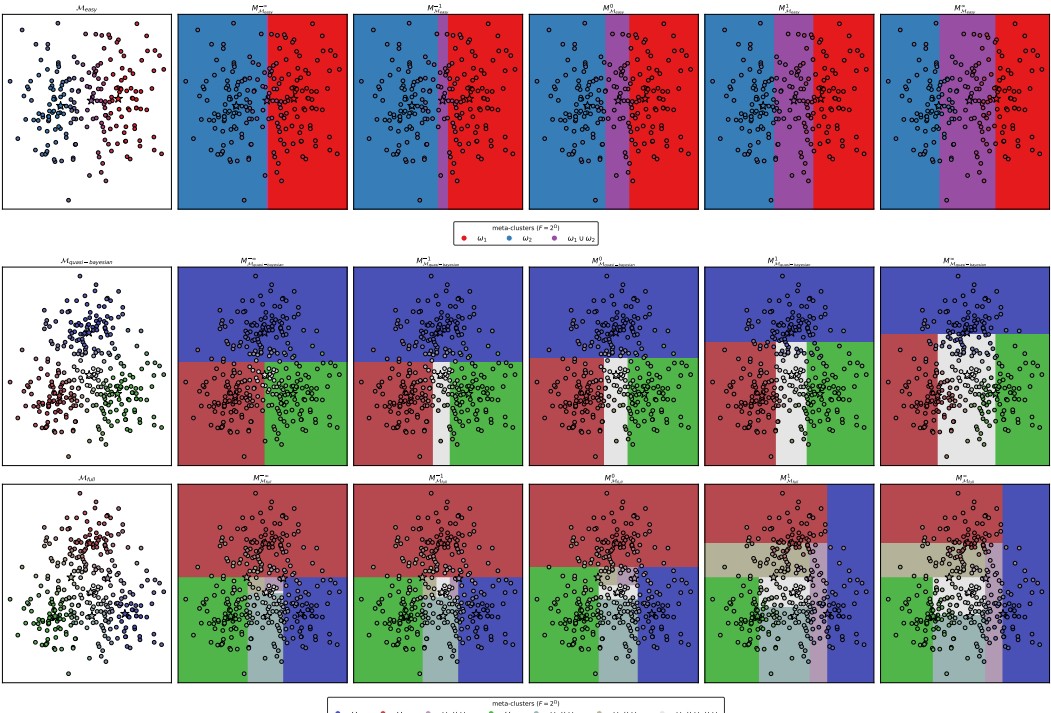

Figure 12: The results of the IEMM on the synthetic dataset for the evidential clustering functions $\mathcal{M}_{\text{easy}}$, $\mathcal{M}_{\text{full}}$, $\mathcal{M}_{\text{quasi-bayesian}}$ and different utility functions. The star represents the centroid of each metacluster. The utility function strongly influences the resulting explanations. The higher the $\lambda$, the more the $\lambda$-evidential mistakeness function assigns larger portions of the space to metaclusters representing doubt. At the limit, $\lambda = -\infty$ (column 2), the obtained explanations give the maximum possible space to the singleton metaclusters. Conversely, $\lambda = \infty$ (column 6) assigns the maximum possible space to the metaclusters representing doubt.

Using the `evclust` library Soubeiga & Antoine (2025), we generated three evidential partitions over synthetic datasets of 2 features ($x$ and $y$). Those were:

- A dataset of 200 entries over which we defined $\mathcal{M}_{\text{easy}}$, with $\Omega = \{\omega_1, \omega_2\}$ and $\mathbb{F}_{\mathcal{M}_{\text{easy}}} = 2^{\Omega} \setminus \varnothing$.

- A dataset of 300 samples and, for $\Omega = \{\omega_1, \omega_2, \omega_3\}$, we generated two types of evidential clustering functions:
  - $\mathcal{M}_{\text{full}}$, an evidential clustering with $\mathbb{F}_{\mathcal{M}_{\text{full}}} = 2^{\Omega} \setminus \varnothing$.
  - $\mathcal{M}_{\text{quasi-bayesian}}$, an evidential clustering that is a quasi-bayesian clustering function. This means that the focal sets are the singletons and the whole space. That is, $\mathbb{F}_{\mathcal{M}_{\text{quasi-bayesian}}} = \{\{\omega_1\}, \{\omega_2\}, \{\omega_3\}, \{\omega_1, \omega_2, \omega_3\}\}$.

Then, for each evidential clustering function, we constructed a decision tree using IEMM and the $\lambda$-evidential mistakeness for different values of $\lambda$. A compact overview of results across utilities and $\lambda$ is provided in Figure 13, while the conflict/non-specificity trends are summarized in Figure 14. The partition of the space induced by these explanations is illustrated in Figure 12. The decision

tree explainer for $\mathcal{M}_{\text{full}}$ and $\lambda = 0$ is shown in Figure 15. The resulting explanations for the quasi-bayesian clustering function are in Table 2.

Table 3 presents the representativeness achieved in each scenario. Notably, the decision tree generated by fixing $\lambda = \infty$ over the $\mathcal{M}_{\text{easy}}$ dataset achieves the highest representativeness observed, surpassing 93%. This can be interpreted as the expected explanation accuracy.

Across both synthetic and real-world datasets, decision trees obtained using the $\lambda$-evidential mistakeness function consistently achieve the best performance in terms of $\mathcal{U}^{\lambda}$-evidential representativeness. The gap between the $\mathcal{U}^{\lambda}$-evidential representativeness and 1 quantifies the loss of accuracy or cost of the explanation.

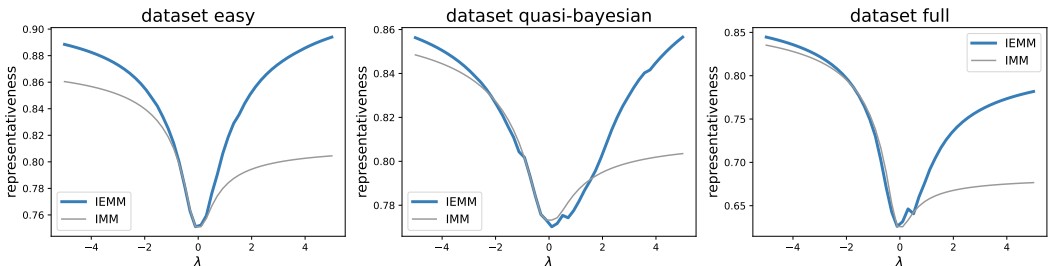

Figure 13: Overview of evidential representativeness across utilities and $\lambda$ on synthetic datasets ($\mathcal{M}_{\text{easy}}$, $\mathcal{M}_{\text{full}}$, and $\mathcal{M}_{\text{quasi-bayesian}}$). The $\lambda$-evidential mistakeness typically yields the best $\mathcal{U}^{\lambda}$-representativeness for its corresponding utility.

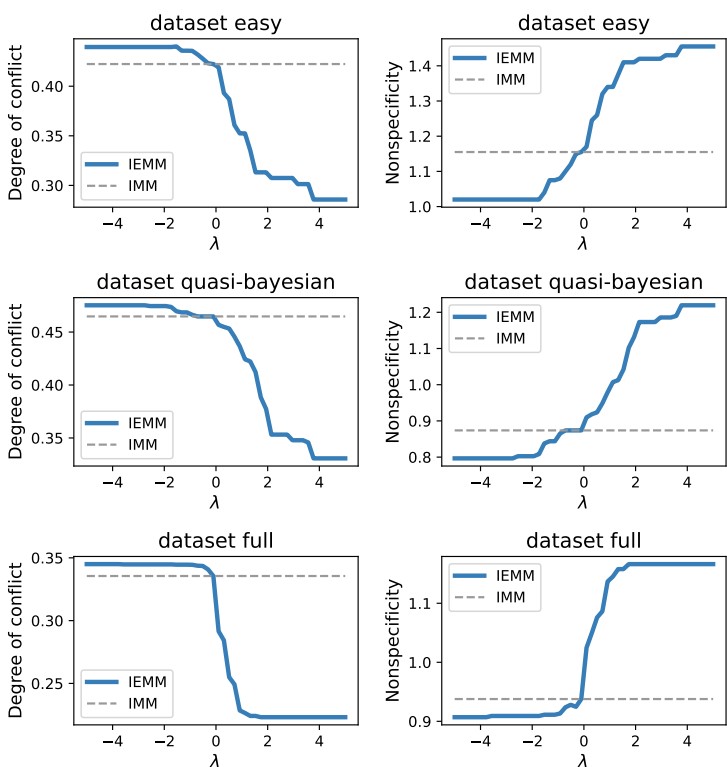

Figure 14: Conflict/non-specificity analysis for the synthetic experiments across $\lambda$ and utilities. As $\lambda$ increases, explanations become more cautious (lower conflict, higher non-specificity), in line with the trade-off in Denoux et al. (2018).

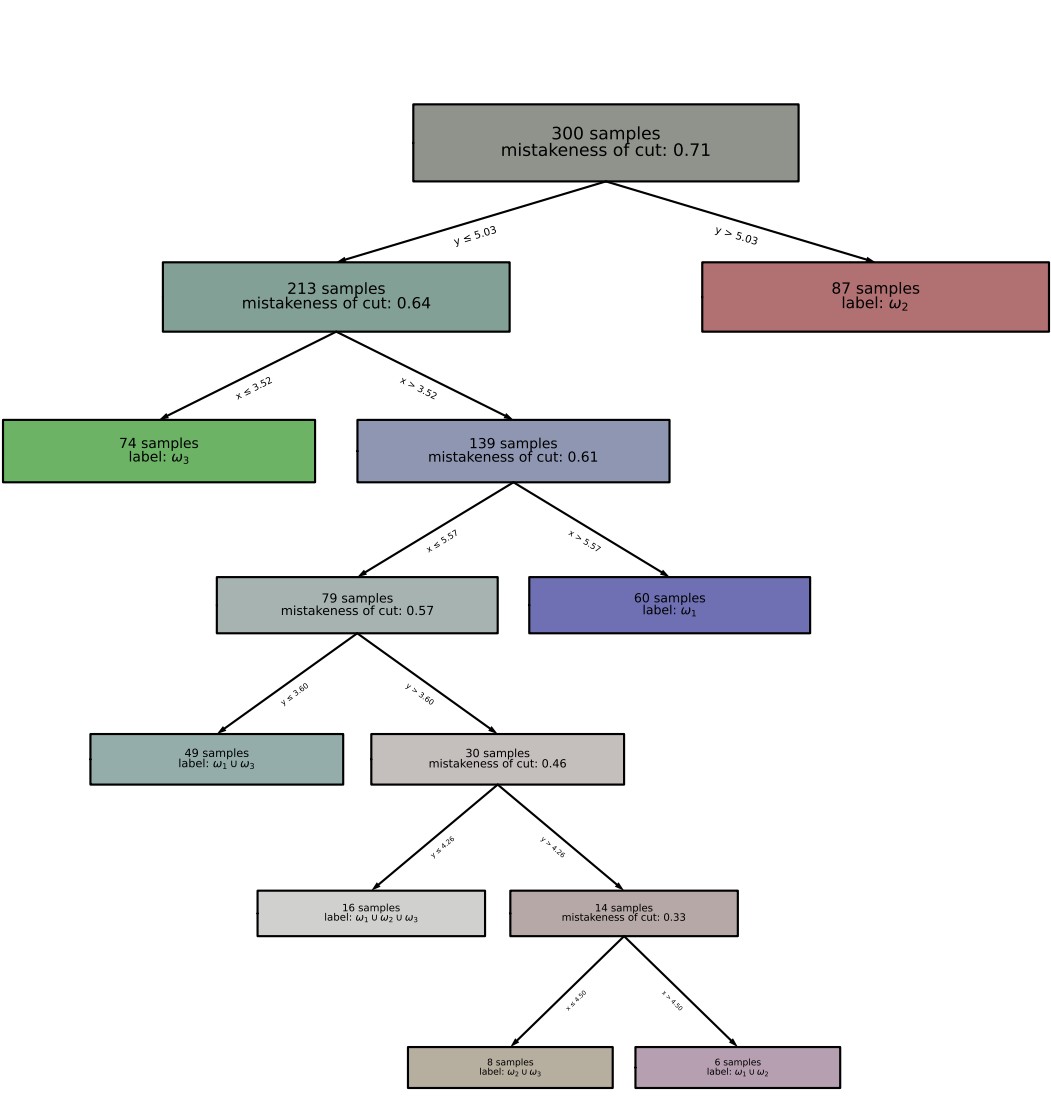

Figure 15: Decision tree obtained with IEMM for the evidential clustering function $\mathcal{M}_{\text{full}}$ and $\lambda = 0$. The decision trees generated by IEMM are shallow by construction, having at most $|\mathbb{F}| - 1$ levels. Each non-terminal node indicates the mistakeness of the corresponding split.

| | $\omega_2$ | $\omega_1 \cup \omega_2 \cup \omega_3$ | $\omega_3$ | $\omega_1$ |
|---|---|---|---|---|
| $M^{-\infty}_{\mathcal{M}_{q-bay}}$ | $(y \leq 4.54) \wedge (x \leq 4.43)$ | $(y \leq 4.54) \wedge (x > 4.43) \wedge (x \leq 4.48)$ | $(y \leq 4.54) \wedge (x > 4.48)$ | $(y > 4.54)$ |
| $M^{-1}_{\mathcal{M}_{q-bay}}$ | $(y \leq 4.54) \wedge (x \leq 4.08)$ | $(y \leq 4.54) \wedge (x > 4.08) \wedge (x \leq 4.98)$ | $(y \leq 4.54) \wedge (x > 4.98)$ | $(y > 4.54)$ |
| $M^{0}_{\mathcal{M}_{q-bay}}$ | $(y \leq 4.69) \wedge (x \leq 3.85)$ | $(y \leq 4.69) \wedge (x > 3.85) \wedge (x \leq 5.09)$ | $(y \leq 4.69) \wedge (x > 5.09)$ | $(y > 4.69)$ |
| $M^{1}_{\mathcal{M}_{q-bay}}$ | $(y \leq 5.39) \wedge (x \leq 3.68)$ | $(y \leq 5.39) \wedge (x \leq 5.28) \wedge (x > 3.68)$ | $(y \leq 5.39) \wedge (x > 5.28)$ | $(y > 5.39)$ |
| $M^{\infty}_{\mathcal{M}_{q-bay}}$ | $(y \leq 5.82) \wedge (x \leq 2.95)$ | $(y \leq 5.82) \wedge (x \leq 5.95) \wedge (x > 2.95)$ | $(y \leq 5.82) \wedge (x > 5.95)$ | $(y > 5.82)$ |

Table 2: Abductive explanations generated by IEMM for all clusters of the quasi-bayesian evidential clustering function. Higher $\lambda$ values result in larger portions of the feature space being attributed to the cautious metacluster $\omega_1 \cup \omega_2 \cup \omega_3$.

| | $\mathcal{R}_{\mathcal{M}_{easy},\mathcal{U}^{-\infty}}$ | $\mathcal{R}_{\mathcal{M}_{easy},\mathcal{U}^{-1}}$ | $\mathcal{R}_{\mathcal{M}_{easy},\mathcal{U}^{0}}$ | $\mathcal{R}_{\mathcal{M}_{easy},\mathcal{U}^{1}}$ | $\mathcal{R}_{\mathcal{M}_{easy},\mathcal{U}^{\infty}}$ |
|---|---|---|---|---|---|
| $M^{-\infty}_{\mathcal{M}_{easy}}$ | **0.915796** | 0.808588 | 0.701380 | 0.701452 | 0.701524 |
| $M^{-1}_{\mathcal{M}_{easy}}$ | 0.901122 | **0.819012** | 0.736903 | 0.749377 | 0.761850 |
| $M^{0}_{\mathcal{M}_{easy}}$ | 0.876733 | 0.813867 | **0.751002** | 0.781731 | 0.812461 |
| $M^{1}_{\mathcal{M}_{easy}}$ | 0.781247 | 0.751198 | 0.721149 | **0.811562** | 0.901975 |
| $M^{\infty}_{\mathcal{M}_{easy}}$ | 0.689432 | 0.669249 | 0.649067 | 0.789613 | **0.930160** |

| | $\mathcal{R}_{\mathcal{M}_{full},\mathcal{U}^{-\infty}}$ | $\mathcal{R}_{\mathcal{M}_{full},\mathcal{U}^{-1}}$ | $\mathcal{R}_{\mathcal{M}_{full},\mathcal{U}^{0}}$ | $\mathcal{R}_{\mathcal{M}_{full},\mathcal{U}^{1}}$ | $\mathcal{R}_{\mathcal{M}_{full},\mathcal{U}^{\infty}}$ |
|---|---|---|---|---|---|
| $M^{-\infty}_{\mathcal{M}_{full}}$ | **0.882009** | 0.731303 | 0.575128 | 0.593766 | 0.612354 |
| $M^{-1}_{\mathcal{M}_{full}}$ | 0.881689 | 0.738726 | 0.598038 | 0.618498 | 0.638218 |
| $M^{0}_{\mathcal{M}_{full}}$ | 0.867413 | **0.745508** | **0.625343** | 0.656206 | 0.683719 |
| $M^{1}_{\mathcal{M}_{full}}$ | 0.642447 | 0.596290 | 0.542490 | **0.681838** | 0.809441 |
| $M^{\infty}_{\mathcal{M}_{full}}$ | 0.621851 | 0.577133 | 0.526137 | 0.679054 | **0.818054** |

| | $\mathcal{R}_{\mathcal{M}_{q-bay},\mathcal{U}^{-\infty}}$ | $\mathcal{R}_{\mathcal{M}_{q-bay},\mathcal{U}^{-1}}$ | $\mathcal{R}_{\mathcal{M}_{q-bay},\mathcal{U}^{0}}$ | $\mathcal{R}_{\mathcal{M}_{q-bay},\mathcal{U}^{1}}$ | $\mathcal{R}_{\mathcal{M}_{q-bay},\mathcal{U}^{\infty}}$ |
|---|---|---|---|---|---|
| $M^{-\infty}_{\mathcal{M}_{q-bay}}$ | **0.887444** | 0.781227 | 0.728118 | 0.728120 | 0.728125 |
| $M^{-1}_{\mathcal{M}_{q-bay}}$ | 0.872286 | 0.799687 | 0.763387 | 0.772136 | 0.789634 |
| $M^{0}_{\mathcal{M}_{q-bay}}$ | 0.866938 | **0.804483** | **0.773256** | **0.785821** | 0.810953 |
| $M^{1}_{\mathcal{M}_{q-bay}}$ | 0.802770 | 0.761888 | 0.741446 | 0.778781 | 0.853452 |
| $M^{\infty}_{\mathcal{M}_{q-bay}}$ | 0.638852 | 0.617576 | 0.606938 | 0.708914 | **0.912866** |

Table 3: Evaluation of the resulting explanations for each metacluster of the synthetic datasets. Each line corresponds to a decision tree trained with one specific mistakeness. Each column corresponds to the $\mathcal{U}$-evidential representativeness of the decision tree. In bold, the best decision tree for each representativeness. We can see that decision trees trained with $\lambda$-evidential mistakeness function tend to be the best in terms of $\mathcal{U}^\lambda$-evidential representativeness.

## I.2 TESTING IEMM AS A CLASSIFIER ON REAL-WORLD DATASETS

To further validate our approach, we also assess IEMM as a stand-alone evidential classifier on larger datasets from the `credal-datasets-master` repository Hoarau et al. (2023b). Unlike explaining a given clustering, this setting requires fitting an evidential model directly from features—learning both decision boundaries and mass assignments—making the task more challenging. We report representativeness in Table 4, provide an overview of per-$\lambda$ behavior in Figure 16, and analyze the conflict/non-specificity trade-off in Figure 17.

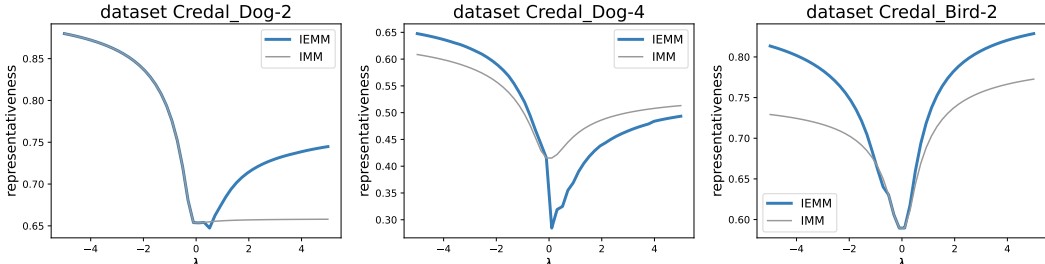

Figure 16: Overview of IEMM classification results across utilities and $\lambda$ on credal datasets. Higher $\lambda$ values lead to more cautious decisions, while lower values favor specificity.

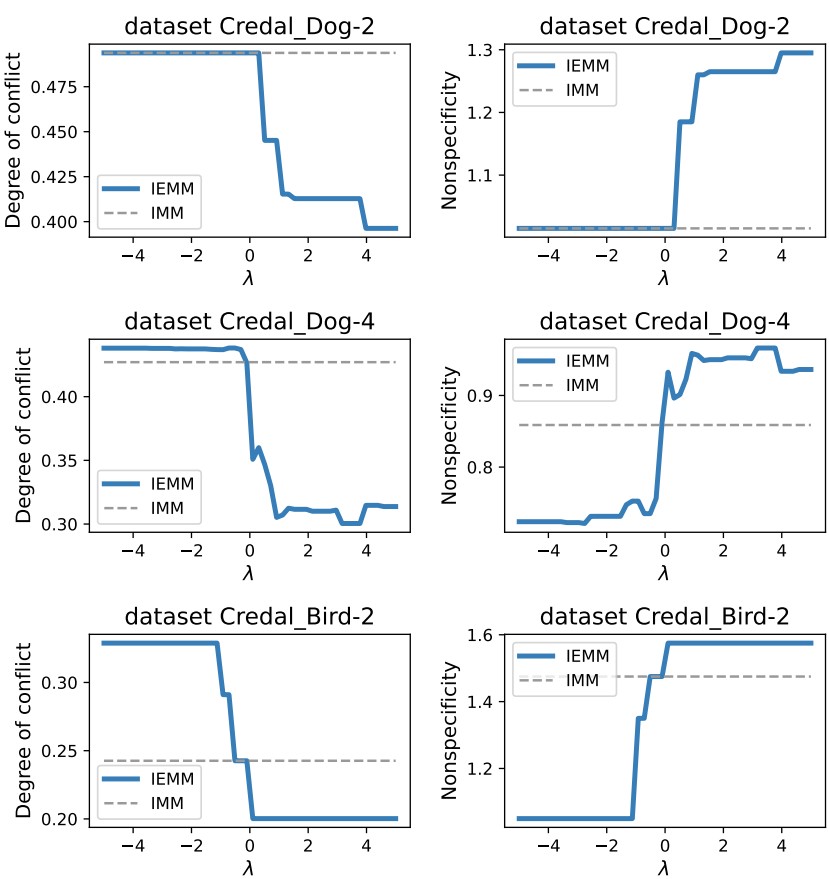

Figure 17: Conflict and non-specificity for IEMM classification across $\lambda$ on credal datasets. The $\lambda$ knob shifts the operating point along the conflict/non-specificity frontier, echoing the behavior observed in clustering.

## I.3 EXTRA RESULTS ON EXPLAINING THE CLUSTERING IN REAL-WORLD DATASETS

Table 5 reports the corresponding representativeness scores on Iris, Wine, and Diabetes.

| | $\mathcal{R}_{\mathcal{M}_{CB-2},\mathcal{U}^{-\infty}}$ | $\mathcal{R}_{\mathcal{M}_{CB-2},\mathcal{U}^{-1}}$ | $\mathcal{R}_{\mathcal{M}_{CB-2},\mathcal{U}^{0}}$ | $\mathcal{R}_{\mathcal{M}_{CB-2},\mathcal{U}^{1}}$ | $\mathcal{R}_{\mathcal{M}_{CB-2},\mathcal{U}^{\infty}}$ |
|---|---|---|---|---|---|
| $M_{\mathcal{M}_{CB-2}}^{-\infty}$ | **0.864286** | 0.658929 | 0.453571 | 0.458929 | 0.464286 |
| $M_{\mathcal{M}_{CB-2}}^{-1}$ | **0.864286** | 0.667857 | 0.471429 | 0.480357 | 0.489286 |
| $M_{\mathcal{M}_{CB-2}}^{0}$ | 0.750000 | **0.669643** | **0.589286** | 0.694643 | 0.800000 |
| $M_{\mathcal{M}_{CB-2}}^{1}$ | 0.714286 | 0.651786 | **0.589286** | **0.726786** | **0.864286** |
| $M_{\mathcal{M}_{CB-2}}^{\infty}$ | 0.714286 | 0.651786 | **0.589286** | **0.726786** | **0.864286** |

| | $\mathcal{R}_{\mathcal{M}_{CD-2},\mathcal{U}^{-\infty}}$ | $\mathcal{R}_{\mathcal{M}_{CD-2},\mathcal{U}^{-1}}$ | $\mathcal{R}_{\mathcal{M}_{CD-2},\mathcal{U}^{0}}$ | $\mathcal{R}_{\mathcal{M}_{CD-2},\mathcal{U}^{1}}$ | $\mathcal{R}_{\mathcal{M}_{CD-2},\mathcal{U}^{\infty}}$ |
|---|---|---|---|---|---|
| $M_{\mathcal{M}_{CD-2}}^{-\infty}$ | **0.913571** | 0.780357 | 0.647143 | 0.647857 | 0.648571 |
| $M_{\mathcal{M}_{CD-2}}^{-1}$ | **0.913571** | **0.783571** | **0.653571** | 0.656071 | 0.658571 |
| $M_{\mathcal{M}_{CD-2}}^{0}$ | **0.913571** | **0.783571** | **0.653571** | 0.656071 | 0.658571 |
| $M_{\mathcal{M}_{CD-2}}^{1}$ | 0.775714 | 0.682500 | 0.589286 | **0.677500** | 0.765714 |
| $M_{\mathcal{M}_{CD-2}}^{\infty}$ | 0.661429 | 0.575000 | 0.488571 | 0.632500 | **0.776429** |

| | $\mathcal{R}_{\mathcal{M}_{CD-4},\mathcal{U}^{-\infty}}$ | $\mathcal{R}_{\mathcal{M}_{CD-4},\mathcal{U}^{-1}}$ | $\mathcal{R}_{\mathcal{M}_{CD-4},\mathcal{U}^{0}}$ | $\mathcal{R}_{\mathcal{M}_{CD-4},\mathcal{U}^{1}}$ | $\mathcal{R}_{\mathcal{M}_{CD-4},\mathcal{U}^{\infty}}$ |
|---|---|---|---|---|---|
| $M_{\mathcal{M}_{CD-4}}^{-\infty}$ | **0.696531** | 0.523750 | 0.416378 | 0.441318 | 0.470153 |
| $M_{\mathcal{M}_{CD-4}}^{-1}$ | 0.694184 | **0.526565** | **0.422959** | 0.454137 | 0.498980 |
| $M_{\mathcal{M}_{CD-4}}^{0}$ | 0.653622 | 0.503031 | 0.414898 | **0.458236** | 0.537194 |
| $M_{\mathcal{M}_{CD-4}}^{1}$ | 0.467959 | 0.345948 | 0.242449 | 0.378104 | 0.543163 |
| $M_{\mathcal{M}_{CD-4}}^{\infty}$ | 0.421224 | 0.320476 | 0.236990 | 0.388116 | **0.601071** |

Table 4: Evaluation of the resulting explanations for each metacluster of the real-world datasets. Each line corresponds to a decision tree trained with one specific mistakeness. We implemented the IEMM algorithm over the datasets Credal_Bird-2 ($\mathcal{M}_{CB-2}$ with 2 classes), Credal_Dog-2 ($\mathcal{M}_{CD-2}$ with 2 classes) and Credal_Dog-4 ($\mathcal{M}_{CD-4}$ with 4 classes). Each column corresponds to the $\mathcal{U}$-evidential representativeness of the decision tree. In bold, the best decision tree for each representativeness. We can see that decision trees trained with $\lambda$-evidential mistakeness function tend to be the best in terms of $\mathcal{U}^{\lambda}$-evidential representativeness.

| | $\mathcal{R}_{\mathcal{M}_{iris},\mathcal{U}^{-\infty}}$ | $\mathcal{R}_{\mathcal{M}_{iris},\mathcal{U}^{-1}}$ | $\mathcal{R}_{\mathcal{M}_{iris},\mathcal{U}^{0}}$ | $\mathcal{R}_{\mathcal{M}_{iris},\mathcal{U}^{1}}$ | $\mathcal{R}_{\mathcal{M}_{iris},\mathcal{U}^{\infty}}$ |
|---|---|---|---|---|---|
| $M_{\mathcal{M}_{iris}}^{-\infty}$ | **0.872336** | 0.746751 | 0.619907 | 0.633296 | 0.646797 |
| $M_{\mathcal{M}_{iris}}^{-1}$ | 0.847368 | 0.776900 | 0.707914 | 0.742323 | 0.777439 |
| $M_{\mathcal{M}_{iris}}^{0}$ | 0.846415 | **0.778853** | **0.712790** | 0.745366 | 0.777935 |
| $M_{\mathcal{M}_{iris}}^{1}$ | 0.773996 | 0.725864 | 0.671799 | **0.749179** | **0.838243** |
| $M_{\mathcal{M}_{iris}}^{\infty}$ | 0.647244 | 0.595757 | 0.538105 | 0.678586 | 0.830750 |

| | $\mathcal{R}_{\mathcal{M}_{wine},\mathcal{U}^{-\infty}}$ | $\mathcal{R}_{\mathcal{M}_{wine},\mathcal{U}^{-1}}$ | $\mathcal{R}_{\mathcal{M}_{wine},\mathcal{U}^{0}}$ | $\mathcal{R}_{\mathcal{M}_{wine},\mathcal{U}^{1}}$ | $\mathcal{R}_{\mathcal{M}_{wine},\mathcal{U}^{\infty}}$ |
|---|---|---|---|---|---|
| $M_{\mathcal{M}_{wine}}^{-\infty}$ | **0.999079** | 0.906952 | 0.814824 | 0.814824 | 0.814824 |
| $M_{\mathcal{M}_{wine}}^{-1}$ | 0.994454 | **0.973001** | 0.951548 | 0.953907 | 0.956266 |
| $M_{\mathcal{M}_{wine}}^{0}$ | 0.986110 | 0.971693 | **0.957276** | 0.963835 | 0.970394 |
| $M_{\mathcal{M}_{wine}}^{1}$ | 0.971606 | 0.963952 | 0.956299 | **0.970139** | 0.983980 |
| $M_{\mathcal{M}_{wine}}^{\infty}$ | 0.623439 | 0.621431 | 0.619423 | 0.807394 | **0.995364** |

| | $\mathcal{R}_{\mathcal{M}_{diabetes},\mathcal{U}^{-\infty}}$ | $\mathcal{R}_{\mathcal{M}_{diabetes},\mathcal{U}^{-1}}$ | $\mathcal{R}_{\mathcal{M}_{diabetes},\mathcal{U}^{0}}$ | $\mathcal{R}_{\mathcal{M}_{diabetes},\mathcal{U}^{1}}$ | $\mathcal{R}_{\mathcal{M}_{diabetes},\mathcal{U}^{\infty}}$ |
|---|---|---|---|---|---|
| $M_{\mathcal{M}_{diabetes}}^{-\infty}$ | **0.854390** | 0.698233 | 0.542076 | 0.542663 | 0.543251 |
| $M_{\mathcal{M}_{diabetes}}^{-1}$ | 0.816186 | **0.701698** | 0.587211 | 0.642283 | 0.697355 |
| $M_{\mathcal{M}_{diabetes}}^{0}$ | 0.783105 | 0.689771 | **0.596438** | 0.675605 | 0.754772 |
| $M_{\mathcal{M}_{diabetes}}^{1}$ | 0.767454 | 0.680228 | 0.593003 | **0.681899** | 0.770795 |
| $M_{\mathcal{M}_{diabetes}}^{\infty}$ | 0.687602 | 0.614695 | 0.541787 | 0.674057 | **0.806328** |

Table 5: Representativeness of IEMM explanations on Iris, Wine, and Diabetes across $\lambda$ values. Higher is better. Best scores per representativeness are highlighted in the tables.

