# OpenReview forum: "Explainable Evidential Clustering"
_ICLR.cc/2026/Conference — Submitted to ICLR 2026_

### Official Review · Reviewer_3JY5 · 2025-10-24

**Soundness:** 2
**Presentation:** 2
**Contribution:** 2
**Rating:** 4
**Confidence:** 3

**Summary:**

The paper introduces Explainable Evidential Clustering, a framework that extends decision-tree–based explanations to clustering models that account for uncertainty. The authors formalise two new theoretical constructs: Evidential Representativeness, a utility-based criterion that quantifies how well an explanation reflects decision-maker preferences, and Evidential Mistakeness, a loss function capturing representativeness errors in evidential settings. The paper provides theoretical analysis, algorithmic formulation, and experiments on small synthetic and real-world datasets.

**Strengths:**

The paper addresses an underexplored problem on explainability for evidential clustering, where model outputs include uncertainty and imprecision. Its main originality lies in adapting the decision-tree explainer paradigm to this evidential setting and introducing formal definitions.

The methodological quality is solid in its theoretical formulation, with proofs, definitions, and algorithmic description. The paper is well written and structured.

In terms of significance, the paper’s contribution is relevant for research on explainability under uncertainty and could serve as a useful foundation for future work in evidential learning. However, its empirical and comparative evaluation is limited, relying on small datasets and a single older baseline, which restricts its practical and experimental impact. Overall, the work is conceptually original and clearly presented, but its experimental and empirical depth remains a key weakness.

**Weaknesses:**

The paper’s main limitation lies in its narrow and weak empirical validation. All experiments are conducted on small, classical datasets (e.g., Iris, Wine, Diabetes, and simple credal datasets), which do not convincingly demonstrate scalability or generalizability. The evaluation includes only one baseline (IMM, 2020), making it difficult to assess whether the proposed method actually advances the state of the art.

The methodological novelty is also limited since IEMM is largely an incremental adaptation of IMM to an evidential setting.

In terms of clarity and framing, while the theoretical exposition is rigorous, the paper occasionally leans too heavily on formalism. It does not provide concrete real-world examples where evidential clustering explanations would offer tangible benefits over simpler alternatives. The related work section is not critical or up-to-date, omitting recent methods for explanations for clustering models.

**Questions:**

1. The related work overview lists some studies but does not critically analyse methodological differences or advances since 2020. Could the authors expand the discussion to include more recent developments (2021–2025) in explainable clustering and evidential learning?
2. Why is the experimental comparison limited to IMM (2020)? Have you considered including other, more recent interpretable clustering baselines?
3. The paper motivates cautiousness in high-stakes domains (e.g., healthcare), but the evaluation does not include such real-world data. Can the authors provide a realistic use case demonstrating IEMM’s interpretability or decision support value?
4. The paper evaluates explanation quality only through the proposed metrics, which measures fidelity to the evidential clustering under a chosen utility. It is not clear why no human-centred or standard explanation quality measures were included. How can one ensure that higher representativeness actually corresponds to more understandable or useful explanations for end users?

---

> ### Author Response · Authors · 2025-11-23
>
> We thank the reviewer for their time and valuable feedback. Addressing their questions:
>
> A1. We have improved our discussion on recent advances in clustering explainability methods, incorporating the reviewer's feedback. We appreciate the suggestion.
>
> A2. We compared our method to an adapted version of IMM to explicitly highlight the impact of the utility function on the final explanation. While the two methods yield similar results under specific conditions, our validation demonstrates how they diverge when those conditions change. It is important to note that adapting IMM involves converting an evidential clustering into a categorical one, which complicates direct comparisons. To maintain a fair and clear evaluation, we focused on this baseline rather than including other IMM-inspired algorithms where this adaptation might introduce further ambiguity.
>
> A3. We thank the reviewer for this suggestion. Following their recommendation, we have added a use case example in Appendix H to demonstrate the value IEMM adds to decision-making. This example shows how IEMM extracts rules from a baseline evidential clustering and adapts them according to the specialist's preferences.
>
> A4. We introduced representativeness as a metric to explicitly capture user preferences regarding imprecision in the explanation. We provide further discussion on this in our response to Reviewer q3cB. Given that explainability for imprecise models is an underexplored field, we are not aware of existing standard quality measures that adequately address this dimension.
>
> We appreciate the reviewer's points and suggestions. We remain open to any further input. If any concerns have not been adequately addressed, we invite the reviewer to reply. We believe the review process has improved the overall quality of our work and look forward to hearing their opinion.

---

### Official Review · Reviewer_Re2y · 2025-10-27

**Soundness:** 3
**Presentation:** 3
**Contribution:** 3
**Rating:** 4
**Confidence:** 3

**Summary:**

This paper proposes a method to generate interpretable, decision-tree-based explanations for evidential clustering. This is a clustering paradigm that models uncertainty and imprecision using Dempster-Shafer theory. The authors introduce new utility-based notions of representativeness and evidential mistakeness, and design the Iterative Evidential Mistakeness Minimization (IEMM) algorithm to greedily optimize decision trees for explaining credal (i.e., evidential) partitions. Theoretical analysis generalizes previous results for hard clustering to the evidential setting and provides utility-based frameworks reflecting different decision-maker attitudes towards caution and error. Experimental validation on synthetic and real-world datasets demonstrates that IEMM delivers more utility-aligned, representative explanations than adapted baselines.

**Strengths:**

1. The paper tackles the largely unexplored challenge of explainability for evidential clustering, a problem that is of interest in high-stakes and risk-averse domains.

2. It reconstructs IMM’s logic for hard cluster explainers, formalizes representativeness in the evidential regime, and proves that minimizing evidential mistakeness yields maximal representativeness under a stakeholder utility.

3. The use of a stakeholder-specific utility to mediate caution versus specificity is conceptually and practically meaningful.

**Weaknesses:**

1. The DSClustering paper (Hovhannisyan, 2025) recently proposed a system that also leverages Dempster–Shafer theory to generate interpretable, rule-based cluster descriptions and to communicate uncertainty to end users. Given this development, the authors of the current submission risk slightly overstating the claim that “no one has addressed interpretability in evidential clustering.” While their approach remains distinct, the paper should avoid implying exclusivity in combining Dempster–Shafer reasoning with interpretability. Instead, the authors should explicitly acknowledge DSClustering as a concurrent but methodologically different effort, emphasizing that their work addresses the post-hoc explanation of existing evidential clusterings.

2. The experiments rely only on small, low-dimensional tabular datasets (Iris, Wine, Diabetes). The absence of high-dimensional, noisy, or real-world examples (e.g., medical or industrial data) limits the demonstration of scalability and generality. Similarly, the baseline (IMM with collapsed labels) cannot express ambiguous metaclusters, making IEMM’s superiority somewhat tautological under the chosen utility. Including stronger baselines, such as CART-style trees trained to maximize expected utility directly, or rule-based interpretable evidential clustering methods, would provide a more compelling comparison.

3. The introduction of a stakeholder-specific utility function U(A,B) is conceptually strong and central to the paper’s stakeholder-aligned framing. However, the paper does not specify how this utility is to be elicited, parameterized, or grounded in user preferences. In practice, different stakeholders, such as clinicians, regulators, or engineers, may have distinct preferences and tolerance levels for ambiguity or misclassification.

4. The exposition in Sections 3–4 is dense and symbol-heavy, which obscures the paper’s otherwise logically theoretical structure. Definitions of representativeness and evidential mistakeness are presented abstractly before any motivating examples, forcing readers to decode complex notation (mass functions, focal sets, utilities) without an intuitive grounding. This could be improved by introducing a running toy example early in the section, visually showing how cautious explanations evolve with different parameters before formal derivations. The paper is overall good and will consider improving the score after clarifications in the rebuttal.

Reference:

Interpretable Clustering Using Dempster–Shafer Theory, Hovhannisyan 2025.

**Questions:**

1. How large can the focal set family |F| become in practice, and do you prune focal sets before running IEMM?

2. Are shallow axis-aligned trees essential for your theoretical guarantees, or could oblique or deeper trees achieve higher representativeness without sacrificing interpretability?

3. How do you envision practical elicitation of the stakeholder utility  𝑈(𝐴, 𝐵)?

---

> ### Author Response · Authors · 2025-11-23
>
> We thank the reviewer for their time and valuable feedback. Addressing their questions:
>
> A1. The focal set's cardinality is never greater than $2^\Omega$. In practical applications, it is sometimes common to restrict the focal sets to singletons and the whole frame $\Omega$, resulting in a cardinality of $|\Omega| + 1$. These restrictions concerning sets are often specified as inputs for the baseline clustering algorithms. Additionally, pruning strategies can be applied by merging focal sets proportionally.
>
> A2. Axis-aligned trees are not strictly essential for the general theoretical guarantees presented in the original paper. In the hard clustering case, explanations using oblique decision trees have been proposed (Gabidolla, 2022). However, following the feedback from Reviewer q3cB, we added a specific guarantee adapted from the IMM framework (appendix G), which does rely on the properties of axis-aligned trees.
>
> A3. Eliciting the utility function in an imprecise setting is indeed a critical and interesting research direction, as noted in the paper. While several parametrization methods have been proposed (Zaffalon, 2012; Yang, 2017; Ma, 2021), there is currently no gold standard in the state of the art.
>
> Regarding the additional comments: We thank the reviewer for bringing the work of Hovhannisyan (2025) to our attention. As it is a very recent publication, we were unaware of it at the time of submission. It is indeed relevant, as it combines three hard clustering outputs using Dempster-Shafer theory. In contrast, our approach generates post-hoc explanations for an existing evidential clustering. We have updated our literature review to include this work and clarify the distinction.
> We also appreciated the suggestion to include a visual toy example alongside the real-world high-stakes medical data to better illustrate the algorithm's utility. We have added such an example to Appendix H, which we believe significantly improves the clarity of the presentation.
>
> We appreciate the reviewer's points and suggestions. We remain open to any further input. If any concerns have not been adequately addressed, we invite the reviewer to reply. We believe the review process has improved the overall quality of our work and look forward to hearing their opinion.
>
> References:
> Gabidolla, M., & Carreira-Perpiñán, M. Á. (2022). Optimal Interpretable Clustering Using Oblique Decision Trees. _Proceedings of the 28th ACM SIGKDD Conference on Knowledge Discovery and Data Mining_, 400–410. [https://doi.org/10.1145/3534678.3539361](https://doi.org/10.1145/3534678.3539361)
> Zaffalon, M., Corani, G., & Mauá, D. (2012). Evaluating credal classifiers by utility-discounted predictive accuracy. _International Journal of Approximate Reasoning_, _53_(8), 1282–1301. [https://doi.org/10.1016/j.ijar.2012.06.022](https://doi.org/10.1016/j.ijar.2012.06.022)
> Yang, G., Destercke, S., & Masson, M.-H. (2017). The Costs of Indeterminacy: How to Determine Them? _IEEE Transactions on Cybernetics_, _47_(12), 4316–4327. [https://doi.org/10.1109/TCYB.2016.2607237](https://doi.org/10.1109/TCYB.2016.2607237)
> Ma, L., & Denœux, T. (2021). Partial classification in the belief function framework. _Knowledge-Based Systems_, _214_, 106742. [https://doi.org/10.1016/j.knosys.2021.106742](https://doi.org/10.1016/j.knosys.2021.106742)

---

### Official Review · Reviewer_uywB · 2025-10-28

**Soundness:** 3
**Presentation:** 3
**Contribution:** 2
**Rating:** 6
**Confidence:** 4

**Summary:**

The paper presents an approach for explaining evidential clustering by training a surrogate decision tree whose leaves are labeled with the
focal elements of the credal partition. To construct the decision tree, the IMM algorithm used for explaining centroid-based methods is
appropriately extended and the Iterative Evidential Mistakeness Minimization (IEMM) method is proposed and evaluated.

**Strengths:**

S1. It seems to be the first approach to explain evidential clustering.
S2. Several novel notions and definitions are included (e.g. cautious explainer)
S3. The proposed IMM extension is well-formulated.

**Weaknesses:**

W1. Evidential clustering has not been widely accepted, especially in real applications
(compared for example to fuzzy or probabilistic clustering methods). Hence an explainer specialized to that framework has limited reach.
W2. If the number of focal elements is large, it seems difficult to interpret the results.

**Questions:**

Q1. Are there approaches that build decision trees to explain fuzzy or probabilistic clustering solutions
taking into account uncertainty in cluster membership? If yes, they should be included in the comparison.
Q2. It would be nice to present the trees obtained for the real datasets considered.

---

> ### Author Response · Authors · 2025-11-23
>
> We thank the reviewer for their time and valuable feedback. Addressing their questions:
>
> A1. We conducted an additional literature search and could not find any existing applications in Fuzzy or Possibilistic methods that generate post-hoc explanations for an initial soft clustering. We have added citations regarding available explainability methods for these other soft clustering approaches to the state-of-the-art overview of our paper.
>
> A2. We have added a new illustrative example in Appendix H, which includes all the generated decision trees. We hope this improves the visualization of the method.
>
> We appreciate the reviewer's points and suggestions. We remain open to any further input. If any concerns have not been adequately addressed, we invite the reviewer to reply. We believe the review process has improved the overall quality of our work and look forward to hearing their opinion.

---

### Official Review · Reviewer_q3cB · 2025-10-28

**Soundness:** 2
**Presentation:** 1
**Contribution:** 2
**Rating:** 2
**Confidence:** 2

**Summary:**

Explainable clustering is a problem that has received considerable theoretical attention in recent years. The setup is the following: Given a dataset and its (hard) cluster assignments, can we find an axis-aligned partition of the data that has competitive performance to the original clustering. This line work started with the IMM algorithm (Moshkovitz et al, 2020), and there has been multiple extensions of the problem setting and methods, but the main focus of this line of work has been on the theoretical guarantees (the so-called price of explainability, and its variations).

The present paper generalises the problem and IMM algorithms to the setting of evidential clustering. In evidential clustering, each data instance is mapped to a probability mass function over the power set of all clusters. This generalises hard clustering, Bayesian clustering (where only singleton clusters have non-zero probabilities), and categorical clustering (where each data is mapped to a single subset of clusters), as illustrated in Figure 5.

In explainable evidential clustering (as posed in the present work), one is given a data set and corresponding mass functions and the goal is to find axis-aligned decision tree based partition of the data such that each data is mapped to a subset of clusters. The authors propose a generalised notion of mistakes, using the mass functions, and extend the IMM algorithm of Moshkovitz et al to obtain an IEMM algorithm, which they empirically evaluate on Gaussian mixtures and few small UCI datasets.

**Strengths:**

The evidential clustering framework is quite interesting, which I believe already provides more interpretability than hard clustering (since the mass function encodes relevant uncertainty in the clusters). The problem of fitting axis-aligned decision trees provides an additional layer of explainability to the problem.

The problem of explainable evidential clustering is also mathematically interesting (at least to those working on explainable clustering) because the criteria from clustering changes from k-means/k-median cost to a more complex setting, where the uncertainty information/mass functions are available.

**Weaknesses:**

The main drawback of the paper is that the problem is not well-formulated. A few important missing pieces are noted below
- The paper proposes IEMM as an approach for explainable evidential clustering without precisely stating the problem (that is, what do we want to minimise while ensuring explainability). One can contrast this with IMM and the corresponding line of work, where the explainable k-means clustering is posed as a problem of achieving low k-means cost while ensuring explainability. As a result IEMM is a heuristic that does not precisely minimise a well-defined loss. This is of the same flavour as hierarchical clustering heuristics, which did not have any sound theoretical basis until Dasgupta's seminal works in 2010s.
- A natural consequence of above is that there are no theoretical guarantees for the proposed IEMM algorithm, which is in contrast to all existing works on explainable clustering that I am aware of. The appendix includes some theoretical results on representativeness, but they do not provide any formal guarantees on the performance of IEMM.
- More fundamentally, the IEMM takes mass functions as input for each data point, but returns only a subset of clusters and not a mass function. Hence, there is loss of information after one approximates the evidential clusters for decision trees (in other words, this method does not seem relevant for explainable Bayesian clustering).
- The experiments are too limited and focus only on simple data, where explainability does not have any consequence. I would be fine with limited experiments if there was strong theoretical contribution. In particular, the explainable clustering literature has been primarily of theoretical interest (with limited use in practice). However, without a strong theory, the paper needs to demonstrate the practical impact of such an approach, where explaining clusters with axis-aligned trees would matter.
- The overall presentation is quite poor, and not written for a general machine learning audience. A large number of concepts are introduced, but not used much in the paper. This makes the paper quite hard to read. I believe the presentation can be significantly simplified by making it more direct. For example, I feel the notion of representativeness is presented in a quite complicated way, making it too formal to even understand how it is useful. The same can be said about the setup, where one can get a better idea only after reading the appendix.

**Questions:**

There are no specific questions. This paper needs to be significantly reworked and rewritten. Please see weaknesses

---

> ### Author Response · Authors · 2025-11-23
>
> We thank the reviewer for their valuable feedback on the importance of theoretical guarantees for IEMM's output, an aspect we initially overlooked. We have now addressed this in the revised paper. Below, we clarify our original approach and the newly added guarantees:
>
> 1. In the original submission, we introduced *representativeness* to quantify how well an explanation aligns with user preferences (cautiousness) encoded in a utility function $\mathcal{U}$. For a categorical evidential clustering $\mathcal{M}\_c$, we proved that a decision tree explanation $L\_A$ for metacluster $A \subseteq \Omega$ achieves maximum representativeness if and only if:
> $$
> \left(\bigwedge\_{\langle\mathcal{A}, v \rangle \in L\_A}\left(x\_\mathcal{A}=v\right)\right) \Rightarrow \mathcal{U}(\overline{\mathcal{M}}\_c(x), A) = 1.
> $$
> Higher $\mathcal{U}$-representativeness indicates that an explanation better reflects the preferences encoded by utility function $\mathcal{U}$.
>
> 2. When the utility function is the indicator $\mathcal{U}(A,B)=1\_{A=B}$, IEMM provides theoretical guarantees on explanation cost. For the Evidential C-means objective function:
> $$
> \mathcal{J}\_{ECM}(\overline{\mathcal{M}}\_c)=\sum\_{x \in X} \left|\overline{\mathcal{M}}\_c(x)\right|^\alpha \left\|x-v\_{\overline{\mathcal{M}}\_c(x)}\right\|^2\_2,
> $$
> we were able to extend guarantees from the IMM framework to IEMM:
> $$
> \mathcal{J}\_{ECM}(\Delta) \leq |\Omega|^\alpha (2 + 8 H |\mathbb{F}\_{\overline{\mathcal{M}}\_c}|) \cdot \mathcal{J}\_{ECM}(\overline{\mathcal{M}}\_c).
> $$
> This proof is now included in **Appendix G** of the revised paper.
>
> 3. IEMM maximizes representativeness (equivalent to minimizing mistakeness by Theorem 2). The reviewer correctly notes the heuristic nature of our algorithm. For general utility functions $\mathcal{U}$ and non-categorical clusterings, exact optimization is intractable, necessitating our heuristic approach. Our experiments demonstrate that this heuristic effectively maximizes representativeness, producing explanations that align better with the specified utility $\mathcal{U}$ compared to baselines using the standard identity utility $\mathcal{U}(A,B)=1\_{A=B}$.
>
> 4. We appreciate the emphasis on clustering cost guarantees, as they ground the quality of IEMM's explanations. The bound we added in appendix G ensures that, for the baseline categorical case with $\mathcal{U}(A,B)=1\_{A=B}$, the explanation cost is bounded relative to the original clustering. For general cases, optimizing specific user preferences, our empirical validation confirms that IEMM successfully improves explanations relative to the chosen utility function in comparison to this baseline.
>
> We thank the reviewer again for their constructive comments and remain open to any further input. If any concerns have not been adequately addressed, we invite the reviewer to reply. We believe the review process has improved the overall quality of our work and look forward to hearing their opinion.

---

> > ### Comment · Reviewer_q3cB · 2025-11-25
> >
> > I appreciate the authors' efforts in providing some theoretical guarantees. However, I am somewhat disappointed that the authors sidestepped the comment by providing guarantees in a rather trivial setting. To be precise, the utility function $U(A,B) = 1_{A=B}$ simply means that we consider a hard clustering problem on the power set of clusters. Hence, the proof of Moshkovitz goes through with cosmetic changes.
> >
> > I am keeping my original score. I believe that a very good series of works can come of this direction, but that would require carefully revisiting the fundamental theoretical questions of explaining evidential clustering in its generality.

---

### Meta-Review · Area_Chair_ki2G · 2026-01-07

**Summary:**

The paper studies post-hoc explainability for evidential clustering.

The main concerns raised by the reviewers were on the lack of empirical validation, lack of compelling presentation with motivating real-world problems or intuitions, a somewhat dense presentation, and a lack of comprehensive comparison/situating with other recent works. Upon studying the authors responses, only some of these aspects were addressed. I think that a major improvement on the presentation, including concrete motivation and intuition as a driving component of the main text (rather than added elements in the appendix) would serve much better a revised version of this paper.

**Reviewer Concerns:**

### Rev q3cB
- The reviewer had issues with the lack of clarity of the formulation/ lack of an optimization objective, and so the algorithm appears as a heuristic. The authors don't contend this, acknowledged the heuristic nature of their approach.

- There's a lack of nontrivial guarantees. The authors added a cost bound in a specific choice of utility function, but the reviewer argues again that this is a trivial result.

- The experimental results is too weak, and the presentation too dense, formal. and difficult to follow.

### Rev uywB
- Notes the limited practical reach, as evidential clustering is a niche problem.
- Missing related work on decision-tree explanations for fuzzy clustering - this is added by the authors upon rebuttal.
- The paper did not contain visualization of the learned trees. The authors provided illustrative examples in the appendix.
- The reviewer notes that the interpretability is decreased when there are many focal elements. The authors mention that pruning can be employed in this cases.

### Rev Re2y
- They note that novelty might be overstated and notes related recent work. The authors added the reference and clarified connections.
- The experiments and baselines are quite weak (small datasets).
- They note that it's not clear how the utilities can be chosen in practice by the user. The authors acknowledge this limitation and comment further on approaches.
- The reviewer also complains about the dense exposition, lack of (timely) intuition and motivating examples. The authors added an illustration/toy example in the appendix.

### Rev 3JY5
- Limited empirical validation noted again.
- Limited novelty (given the work on IMM) and outdated related work. The authors expanded the discussions on recent works to partially address this.
- Lack of real-world use cases: the authors added an illustrative decision-support example in Appendix H.

**Reviewer Scores:**

- Rev q3cB (2, conf 2): The reviewer is not convinced by the authors' reply, unlikely to change scores. However, confidence is very limited.
- Rev uywB (6, conf 4): The reviewer did not engage further, and the comments were partially resolved.
- Rev Re2y (4, conf 3): some of the questions were answered, however unlikely to change scores given the limited experiments and dense presentation.
- Rev 3JY5 (4, conf 3): unlikely to change scores given the lack of motivating examples and limited empirical validation.

---

### Decision · Program_Chairs · 2026-01-26

Reject